

# Combining the U-Net model and a Multi-textRG algorithm for fine SAR ice-water classification

Yan Sun[1,2], Shaoyin Wang[1,2], Xiao Cheng[1,2], Teng Li[1,2], Chong Liu[1,2], Yufang Ye[1,2], Xi Zhao[1,2]

[1]School of Geospatial Engineering and Science, Sun Yat-sen University, Guangzhou, China

[2]Southern Marine Science and Engineering Guangdong Laboratory and the Key Laboratory of Comprehensive Observation of Polar Environment, Ministry of Education, Sun Yat-sen University, Zhuhai, China

*Correspondence to*: Xiao Cheng (chengxiao9@mail.sysu.edu.cn)

**Abstract.** Synthetic Aperture Radar (SAR)-based sea ice classification faces challenges due to the similarity among surfaces such as wind-driven open water (OW), smooth thin ice, and melted ice surfaces. Previous algorithms combine pixel-based

and region-based machine learning methods or statistical classifiers, yet struggle with hardly improved accuracy arrested by the fuzzy surfaces and limited manual labels. In this study, we propose an automated algorithm framework by combining the semantic segmentation of ice regions and the multi-stage detection of ice pixels to produce high-accuracy and high-resolution ice-water classification data. Firstly, we used the U-Net convolutional neural networks model with the well processed GCOM-W1 AMSR2 36.5GHz H polarization, Sentinel-1 SAR EW dual-polarization data, and CIS/DMI ice chart

labels as data inputs to train and perform semantic segmentation of major ice distribution regions with near-100 % accuracy. Subsequently, within the U-Net semantically segmented ice region, we redesigned the GLCM textures and the HV/HH polarization ratio of Sentinel-1 SAR images to create a combined texture, which served as the basis for the Multi-textRG algorithm to employ multi-stage region growing for retrieving ice pixel details. We validated the SAR classification results on Landsat-8 and Sentinel-2 optical data yielding an overall accuracy (OA) of 84.9 %, a low false negative (FN) of 4.24 %

indicating underestimated low backscatter ice surfaces, and a higher false positive (FP) of 10.8 % reflecting their resolution difference along ice edges. Through detailed analyses and discussions of classification results under the similar ice and water conditions mentioned at the beginning, we anticipate that the proposed algorithm framework successfully addresses accurate ice-water classification across all seasons and enhances the labelling process for ice pixel samples.

## 1 Introduction

Precise sea ice observations, particularly during the melting season, are critical for studying local ocean-ice-atmosphere interactions. Currently, sea ice monitoring using synthetic aperture radar (SAR) data faces significant limitations in algorithm accuracy and robustness. SAR-based sea ice classification algorithms have evolved from intensity threshold segmentation to advanced convolutional neural network (CNN) methods. As powerful tools for image processing, CNNs have demonstrated great potential in SAR-based sea ice classification (Leigh et al., 2014; Kortum et al., 2021; Chen et al.,



2023). However, achieving high-resolution and high-accuracy predictions with CNNs requires a sufficiently large dataset of fine-scale sample labels, which is largely unavailable even for the hand drawn.

This study proposed an algorithm framework that combines CNNs and the Multi-textRG algorithm, a multi-layer GLCM texture-based regional growing method, to enable automated or semi-automated sample labelling. The framework leverages existing coarsely labelled ice charts to train a U-Net CNN model for semantic segmentation of major ice regions

and wind-driven open water, one of the surface types that are easily misidentified. Subsequently, we integrated expert classification knowledge into the Multi-textRG algorithm to achieve detailed recognition of ice pixels within the U-Net segmented ice regions. The Multi-textRG algorithm demonstrates high accuracy and robustness when the U-Net model performs the semantic segmentation with high precision. Achieving automated ice-water sample labelling means achieving accurate and robust automated ice-water classification.

Studying on previous SAR-based sea ice classification algorithms, except for complete CNN model training, we believe the algorithm procedure of combining CNN for semantic segmentation and then empirical methods for details recognition is the optimal automatic labelling approach. Similar with the reviews in (Zakhvatkina et al., 2019) and (Song et al., 2021), we categorize previous SAR-based sea ice classification algorithms into four groups. 1), pixel-based machine learning methods utilize image textures for classification, including SVM, RF, CRF and Multilayer Perceptron (MLP) methods that are trained

on gray level co-occurrence matrix (GLCM) textures. These methods primarily rely on local texture features, making them highly sensitive to pixel-level details (Liu et al., 2015; Murashkin et al., 2018; Park et al., 2020; Zakhvatkina et al., 2017; Zakhvatkina et al., 2013). 2), polygon-based image segmentation is combined with pixel-based machine learning, such as the iterative region growing using semantics (IRGS)-SVM (i.e., MAGIC system) method (Leigh et al., 2014) and the IRGS-RF method (Jiang et al., 2022). The IRGS image segmentation and machine learning are conducted separately, and the final

ice/water labels are determined by incorporating an energy function (Leigh et al., 2014). One attempt at super-high resolution ice-water segmentation is by (Korosov and Jeong-Won, 2016). The authors first use K-means segmentation in each 300×300 pixels block to obtain finely segmented small polygons of the whole Sentinel-1 SAR image, and then calculate the GLCM textures in each polygon (one polygon as one texture window) as the input to training SVM using manual labels on these polygons. 3), statistical models are used as classifiers, like MSTA-CRF (Zhang et al., 2021), CRF

combined with CNN (Kortum et al., 2021) and clustering algorithms accounting for textural variations (Cristea et al., 2022; Cristea et al., 2020; Doulgeris, 2015). 4), the fourth category features polygon-based deep learning models directly training on SAR pixel intensities, employing convolutional structures to retrieve semantic context information (Boulze et al., 2020; Malmgren-Hansen et al., 2021; Song et al., 2021; Song et al., 2018; Wang and Li, 2020). Deep learning models are also used for SAR sea ice concentration (SIC) retrieval using Canadian Ice Service (CIS) charts or Advanced Microwave Scanning

Radiometer 2 (AMSR2) SIC products as labels (Cooke and Scott, 2019; Wang et al., 2017a; Wang et al., 2017b; Wang et al., 2016). Kucik and Stokholm (2023) reviewed the CNN algorithms developed on SAR images to estimate SIC or ice types.

For the second group of image clustering segmentation combined with machine learning and the fourth group of CNN models relying on sample labelling, the high similarity between several ice types is the major error source. Smooth thin ice



(including newly formed ice, level young ice, and level first-year ice) and wetted ice (such as melt ponds and wet snow covered ice), along with calm open water, generally exhibit extremely low backscatter values (Cristea et al., 2022; Niehaus et al., 2023; Song et al., 2021; Zakhvatkina et al., 2017). Except for the linear features of newly formed ice in ice leads, smooth thin ice and wetted ice typically have similar regional semantic context and texture features to open water areas. For instance, bright young ice and heavily deformed ice (HDefI), as well as dark young ice and deformed ice (DefI), show highly overlapping SAR backscatter and GLCM texture histogram distributions (Guo et al., 2023). Newly formed ice in disk-shaped melt ponds not only has extremely low backscatter but also creates high-contrast boundaries with surrounding ice, particularly in Sentinel-1 SAR HV images, leading to largely potential misclassification as open water (Sun et al., 2023). Additionally, winded open water could have significantly higher backscatter and texture values than smooth thin ice, even reaching the levels of thick multi-year ice (Boulze et al., 2020; Li et al., 2021; Song et al., 2021). Various surface characteristics (such as different degrees of deformation, melting, geometric roughness, or internal micro-roughness, salinity, and wind speed) and radar parameters (such as wavelength, polarization, and incidence angle) are crucial factors influencing SAR backscatter echoes (Guo et al., 2023; Lohse et al., 2020; Song et al., 2021). Even manually labelled samples, e.g., ice charts, can also be influenced by the high inter-class ambiguity (Kortum et al., 2021; Park et al., 2020; Song et al., 2021). Moreover, since ice charts are created by integrating multiple data sources and serve polar navigation, they typically overestimate sea ice extent by erroneously including calm open water within the ice boundaries.

CNN-based classification requires semantic information within a receipt field of at least 500 pixels of 80 m resolution to achieve good performance (Stokholm et al., 2022). Therefore, when the image clustering and machine learning focus on small-scale and high-resolution classification, the second group of algorithms could introduce double segmentation errors on these similar surface types. Whether using texture features or convolutional operations, machine learning-based methods in all four algorithm groups leverage semantic information from various window or image block sizes. Texture-based classification typically trains on single-point texture values from windows with tens of pixels, which often fail to capture effective identification for the above-mentioned similar surfaces. However, using larger semantic contexts inevitably sacrifices the original pixel resolution. Combining region-based and pixel-based image features is the only effective way to achieve precise and detailed ice-water classification, which requires both the improvement of CNN models and fine-scale sample labels.

In this paper, the proposed algorithm framework beginning with a region-based CNN semantic segmentation and followed by texture-pixel-level Multi-textRG image segmentation developed on experiences and statistics. The AI4Arctic project provides pre-processed Sentinel-1 SAR data, GCOM-W1 AMSR2 brightness temperature data, and a pre-trained U-Net CNN model. Thus, we utilize the AI4Arctic project dataset and the U-Net model to achieve semantic segmentation of major ice regions, and then use raw Sentinel-1 SAR data with completely thermal noise removal to calculate GLCM textures for regional growing of ice pixels. In the following, Section 2 introduces the input data for the algorithm framework and the independent validation data for classification results. Section 3 illustrates the whole framework/workflow including the U-Net CNN and the Multi-textRG algorithm. The results of U-Net predicted ice region, Multi-textRG detected ice and water



pixels, and their validations to Landsat-8 or Sentinel-2 optical data are displayed in Section 4. Sequentially, Sections 5 and 6 give out the discussions and conclusions of this paper.

## 2 Data

### 2.1 AI4Arctic project dataset

AI4Arctic project, also called AutoICE challenge, aims to advance CNN-based sea ice parameter retrieval from SAR data with increased robustness and accuracy. The AI4Arctic project datasets focus primarily on SAR data, supplemented with AMSR2 data and ERA5 weather data, which possess numerous advantages for SAR-based sea ice monitoring studies. The SAR data were selected based on expert experience, featuring typical classification challenges of ice and water surface characteristics while maintaining a certain feature balance.

Within the ready-to-train dataset, the Sentinel-1 dual-polarization EW mode SAR data, full-band (seven bands, two polarizations) AMSR2 Level-1B brightness temperature (TB) data, ERA5 meteorological data, as well as Canadian Ice Service (CIS) and Danish Meteorological Institute (DMI) ice chart data are well pre-processed. The latest version of the dataset, version 2, was released on May 25, 2023, and includes 512 training and 20 test files (https://platform.ai4eo.eu/auto-ice/data). The Sentinel-1 SAR data were denoised with the method developed by the Nansen Environmental and Remote Sensing Center (NERSC) (Korosov et al., 2022). The SIC in CIS/DMI ice charts were conversed from WMO Egg code to 11 classes (0 to 10, 10% interval). In the ready-to-train dataset, the SAR HH and HV backscatters, incidence angle, distance-to-land map and the ice charts SIC are all resampled into 80 m pixel space resolution, with a matrix size of around 5000×5000. The other data are all resampled into 2 km grid resolution, co-located and georeferenced to the geometries of Sentinel-1 SAR images. Except for ice charts, the other data were all standard normalized providing with normalization parameters. For more information, please refer to the AI4Arctic project dataset user manual. Relevant papers associated with the AI4Arctic project include (Esbensen, 2022; Kucik and Stokholm, 2023; Stokholm et al., 2023; Stokholm et al., 2022).

The AI4Arctic project also provides a pre-trained U-Net CNN model for predicting sea ice concentration, facilitating the fine-tuning of the U-Net model in the first step of our algorithm framework. The AI4Arctic dataset mainly covers the waters around Greenland and the Canadian Arctic Archipelago (see Section 2.5), where optical satellites, e.g., the Landsat series and Sentinel-2, provide valuable observations and offer the possibility of independent validation. In this paper, we utilize the ready-to-train dataset and U-Net model provided by the AI4Arctic project to achieve semantic segmentation of ice regions in SAR images.

### 2.2 Sentinel-1 SAR data

We use dual-polarization (HH, HV) EW mode Level-1 GRD data provided by the Sentinel-1A and 1B satellites operated by the European Space Agency (ESA). Each satellite is equipped with an advanced C-band (5.405 GHz) SAR sensor, capable



of providing continuous all-weather, day-and-night imaging of the Earth's surface. The EW mode observation images have an incidence angle ranging from 20° to 45°, a swath width of 400 km, and a spatial resolution of 20 m × 40 m.

The complete processing of Sentinel-1 dual-polarization data includes latitude-longitude interpolation, land masking, calibration, thermal noise removal, and additional incidence angle correction for HH images. The authors of the AI4Arctic project have preprocessed and normalized the Sentinel-1 dual-polarization data to facilitate the training of neural network models. Specifically, the AI4Arctic dataset uses the state-of-the-art NERSC algorithm (Korosov et al., 2022) to correct the thermal noise in Sentinel-1 HV polarization images. Based on the method proposed by (Park et al., 2018), the NERSC

algorithm enhances the thermal noise removal procedure by adjusting the shape of the annotated range noise vectors provided by ESA to the shape proportional to the antenna range gain $G_{ar}$. However, the AI4Arctic provided SAR data only removed thermal noise, leaving texture noise (also known as multiplicative thermal noise) existing. On calm open water and level ice surfaces without frost flowers, texture noise can produce GLCM texture features similar to those of higher backscatter ice (Park et al., 2019). This can significantly interfere with the Multi-textRG algorithm, which uses GLCM

textures for fine-grained sea ice pixel identification.

Therefore, we use the thermal noise removal algorithm combining that proposed by (Sun and Li, 2021) and by (Park et al., 2019) to newly calibrate and denoise the raw Sentinel-1 SAR data, which were downloaded from the ESA website. The public codes can be accessed at https://zenodo.org/records/10997427 or https://zenodo.org/records/13269639.

**2.3 GCOM-W1 AMSR2 data**

We use the AMSR2 Level-1B 36.5 GHz H-polarization TB data, provided by the GCOM-W1 satellite launched by the Japan Aerospace Exploration Agency (JAXA). AMSR2 is a passive microwave (PM) instrument specifically designed to monitor microwave radiation from the Earth's surface and atmosphere. AMSR2 data can be acquired both day and night, with a spatial coverage exceeding 1450 km and an orbital period of approximately 100 minutes. The AMSR2 can image at frequency bands of 6.925/7.3, 10.65, 18.7, 23.8, 36.5, and 89.0 GHz, and in both H and V polarizations. The spatial

resolution of AMSR2 radiometric data improves from 35 km × 62 km at low-frequency bands to 3 km × 5 km at high-frequency bands.

As illustrated in the channel specifications (https://suzaku.eorc.jaxa.jp/GCOM_W/wamsr2/whatsamsr2.html), the 23.8 GHz and lower frequency bands not only have low spatial resolutions but also high sensitivity to sea surface temperature or integrated water vapor. The 89.0 GHz band offers the highest spatial resolution and simultaneous high sensitivity to sea

surface temperature, integrated water vapor, integrated cloud liquid water, and sea surface wind surface. Comparatively, the 36.5 GHz band has relatively lower sensitivity to environmental factors and relatively high spatial resolution. The algorithm fusing SAR with AMSR2 data aims to distinguish between ice/sea surfaces with different wind speeds and temperatures. Therefore, we select the AMSR2 36.5 GHz H-polarization TB data for combined input.



**2.4 Landsat-8 and Sentinel-2 validation data**

Landsat-8 satellite and the Copernicus Sentinel-2A, 2B constellation are two important optical imaging systems. Landsat-8 level-2 and Sentinel-2 level-1C data both provide top-of-atmosphere (TOA) reflectance in the spectral bands of visible, near-infrared (NIR) and shortwave infrared (SWIR) ranges. Additionally, the two satellites provide a quality assessment (QA) band including snow/ice signs and cloud signs. The QA snow/ice signs in Landsat-8 and Sentinel-2 data are generated by strict cloud and snow detection algorithm sequences, see that of the Sentinel-2 Level-1C/2A product (https://sentinels.

copernicus.eu/web/sentinel/technical-guides/sentinel-2-msi/level-2a/algorithm-overview). Thus, we directly used the QA snow/ice signs excluding lands as "true" sea ice observations to validate our ice-water classification. We also used the TOA reflectance in natural color bands (red, green, blue) to visualize the optical features of sea surfaces, and the NIR and SWIR bands to manually correct the snow/ice detection.

The Landsat-8 and the Sentinel-2 data were respectively selected from the collection 'LANDSAT/LC08/C02/T1_L2'

and the collection 'COPERNICUS/S2_SR' overlapping the geographic location of each SAR scene and within 1 day before or after the SAR imaging-time, merged by median calculation, and cloud-masked by using the QA cloud signs on the Google Earth Engine (GEE) platform. Then, the natural color bands, NIR, SWIR, and QA snow/ice signs (total 6 bands) of both Landsat-8 and Sentinel-2 are exported from the GEE platform by blocks with a resolution of 30 m. The block size is determined to improve export speeds on the GEE platform. All exported bands are resampled to SAR image geolocations.

Evaluation metrics are then calculated based on pixel-to-pixel validation, and the visualized comparisons for 219 matched images are provided in "validation-to-L8S2.zip" available at https://zenodo.org/records/13269639.

Surfaces covered by clouds could be mistakenly identified as snow/ice or open water due to errors in the QA cloud signs of Landsat-8 and Sentinel-2 data. Therefore, after the initial validation and visualization, we selected images with cloud-interfered errors to conduct further correction using the Modified Normalized Difference Water Index (MNDWI) (Xu,

2007). The equation is given below:

$$MNDWI = \frac{Green - SWIR}{Green + SWIR} \tag{1}$$

Here, MNDWI is calculated by normalizing the difference between the green band (Band 3 in Landsat-8 and Sentinel-2) and the SWIR band (Band 7 in Landsat-8: 2.107–2.294 μm; Band 12 in Sentinel-2: 2202.4, 2185.7 nm) over their sum. Based on experiments, MNDWI shows significant potential in distinguishing open water under thin/cirrus cloud cover from

sea ice, a capability that other indices lack. Typically, MNDWI yields very low values for open water under thin/cirrus clouds, but it can also produce extremely low values on certain sea ice surfaces. As a result, MNDWI does not have a consistent value distribution or a stable segmentation threshold for ice-water classification. The threshold of 0.75 is a reference value. In this paper, we select the MNDWI threshold through visual interpretation to identify open water under thin/cirrus cloud cover, to correct the QA snow/ice sign.



The pixel-based validation metrics between SAR detected ice result and Landsat-8/Sentinel-2 QA snow/ice sign include overall accuracy (OA: sum of true positive (TP) and true negative (TN)), false negative (FN: ice in Landsat-8/Sentinel-2 and OW in SAR) rate, false positive (FP: OW in Landsat-8/Sentinel-2 and ice in SAR) rate, and $Parea$ (area proportion of effective pixels). They are given below:

$$OA = \frac{\{ICE_{l8s2} \; and \; ICE_{sar}\}_{np}}{notnan_{np}} + \frac{\{OW_{l8s2} \; and \; OW_{sar}\}_{np}}{notnan_{np}} \tag{2}$$

$$FN = \frac{\{ICE_{l8s2} \; and \; OW_{sar}\}_{np}}{notnan_{np}} \tag{3}$$

$$FP = \frac{\{OW_{l8s2} \; and \; ICE_{sar}\}_{np}}{notnan_{np}} \tag{4}$$

$$Parea = \frac{notnan_{np}}{notnan_{np}+nan_{np}} \tag{5}$$

The subscript $np$ is the number of pixels, $notnan_{np}$ is the number of valid pixels in both the SAR and Landsat-8/Sentinel-2 images, and $nan_{np}$ represents the number of pixels masked as NaN values (including land masks and cloud masks) in either the SAR or Landsat-8/Sentinel-2 images. Thus, $Parea$ is used as the weights for calculating the average OA values among totally 219 SAR images with matched Landsat-8/Sentinel-2 images. Then $ICE_*$ and $OW_*$ respectively denote ice pixels and open water pixels, where the subscript $sar$ means the Multi-textRG ice-water classification in SAR images, and the subscript $l8s2$ means the QA snow/ice sign in Landsat-8/Sentinel-2 images.

**2.5 Data coverage**

The Sentinel-1 SAR images in the AI4Arctic dataset cover the waters surrounding Greenland and the Canadian Arctic Archipelago from January 8, 2018, to December 21, 2021. These regions are characterized by rapid ice drift, continuous and fast-changing freeze or melt processes throughout the year, high wind speeds over open water, and frequent offshore wind streaks, all of which pose significant challenges for SAR-based sea ice classification algorithms. Certain challenging types, even for professional ice analysts, are difficult to accurately interpret during ice chart production (Stokholm et al., 2022). This difficulty may be attributed to ambiguous surface features (e.g., ice covered by melt ponds, smooth first-year ice, deformed ice, new ice, and thin ice). As shown in Fig. 1, the SAR images exhibit uniform temporal and spatial distribution patterns. Therefore, the AI4Arctic dataset is highly valuable.

In Fig. 1(a), the northernmost point imaged by Landsat-8 and Sentinel-2 optical satellites reaches the northern coast of Greenland. However, the number of usable scenes is significantly limited due to widespread cloud contamination. As shown in Fig. 1(c) and (d), a total of 85 SAR images matched with valid Landsat-8 data after cloud masking; a total of 134 SAR images matched with valid Sentinel-2 data (launched in February 2019) after cloud masking; of them, 54 SAR images had



matching data from both Landsat-8 and Sentinel-2. The optical images from the two satellites are densely distributed during the melt season and early freeze-up season.

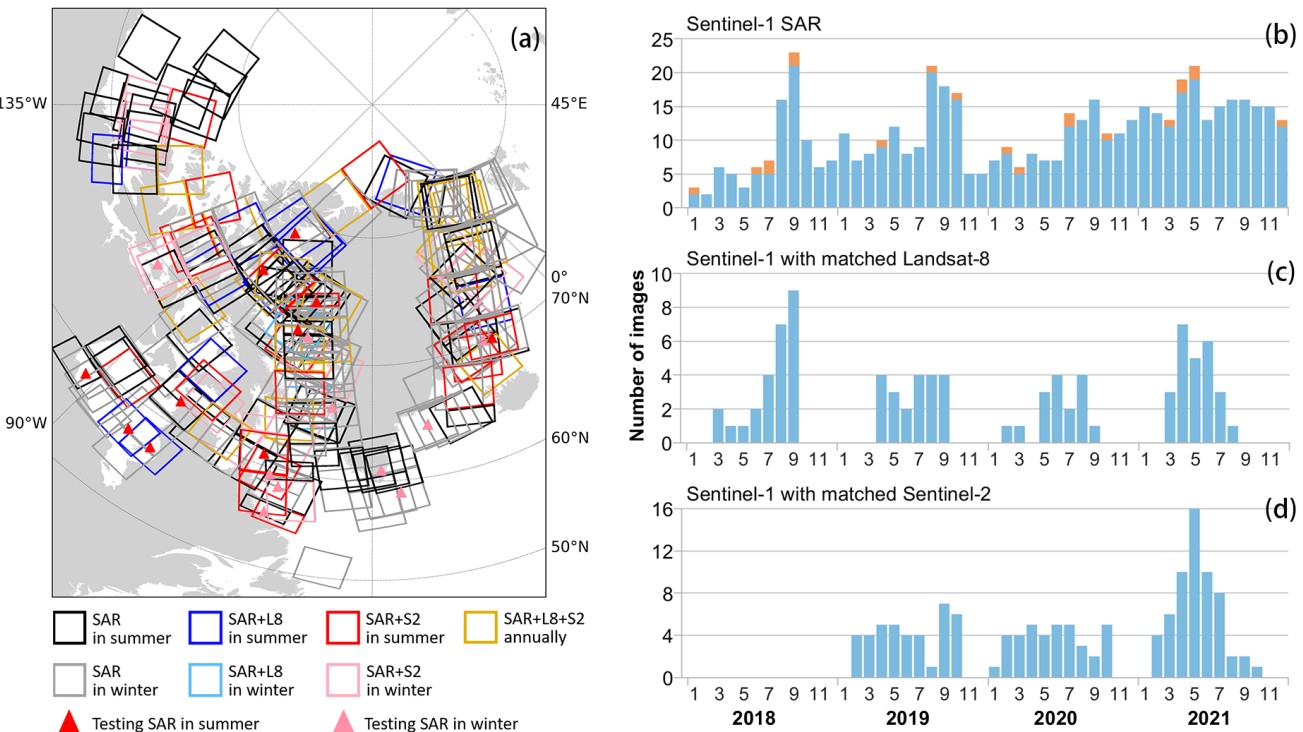

**Figure 1.** Data distribution. (a) shows the geolocations of 512 training images (boxes) and 20 testing images (triangles), which are differentiated with matched Landsat-8/Sentinel-2 (L8/S2) data or not, and in summer (May to September) or winter (other months). (b) is the number distribution of all Sentinel-1 SAR images (blue: training, orange: testing) in month of the year 2018 to 2021. (c) and (d) respectively shows the number distribution of Sentinel-1 SAR images with matched Landsat-8 (85 scenes) and Sentinel-2 (134 scenes) images.

## 3 Methods

To integrate region-level and pixel-level ice detection, we first employed a U-Net CNN model trained on coarsely labelled ice charts to extract the overall ice distribution domain. The objective was to enable the U-Net model to learn sufficient semantic context within a larger convolution field to distinguish wind-driven open water from smooth thin ice, wetted ice and other ice surfaces with similar characteristics in high accuracy. Within the U-Net segmented ice regions, the Multi-textRG algorithm performs multi-layer ice pixel recognition based on feature/metric threshold segmentation, similar to the cloud mask procedures used in optical imagery. Building on previous studies, GLCM texture features and the HV/HH ratio from Sentinel-1 SAR images were utilized to create a combined texture feature with relatively high and stable ice-water contrast under various ice conditions. This combined feature served as the foundation for the region growing algorithm, which extracts "ice seeds" based on local intensity thresholds and selects (or "grows") new ice seeds using local intensity



difference thresholds. Section 3.1 and 3.2 provide the technical details of the U-Net experimental settings and the Multi-textRG algorithm procedure, respectively.

### 3.1 The U-Net model

#### 3.1.1 Experimental settings

In this paper, we utilized the U-Net CNN model provided by the AI4Arctic project. The effects of different receptive
fields (RFs), loss functions, and optimization objectives based on regression or classification on the performance of the U-Net model have been deeply explored (Kucik and Stokholm, 2023; Stokholm et al., 2023; Stokholm et al., 2022). The model structure can refer to Fig. 5 in (Stokholm et al., 2022). Figure 2 shows the example input and output of the simplified U-Net model. The normalized AMSR2 36.5GHz H-polarization data, Sentinel-1 SAR HH and HV polarization data, Sentinel-1 SAR incidence angle, and distance to land map, five features in total, were used as inputs to the model. The SIC ice chart
produced by DMI/CIS is used as sample labels for the model. During the U-Net model training, both the model inputs and sample labels were resampled to an 80 m pixel spacing. Table 1 lists the experimental parameters for the model setup.

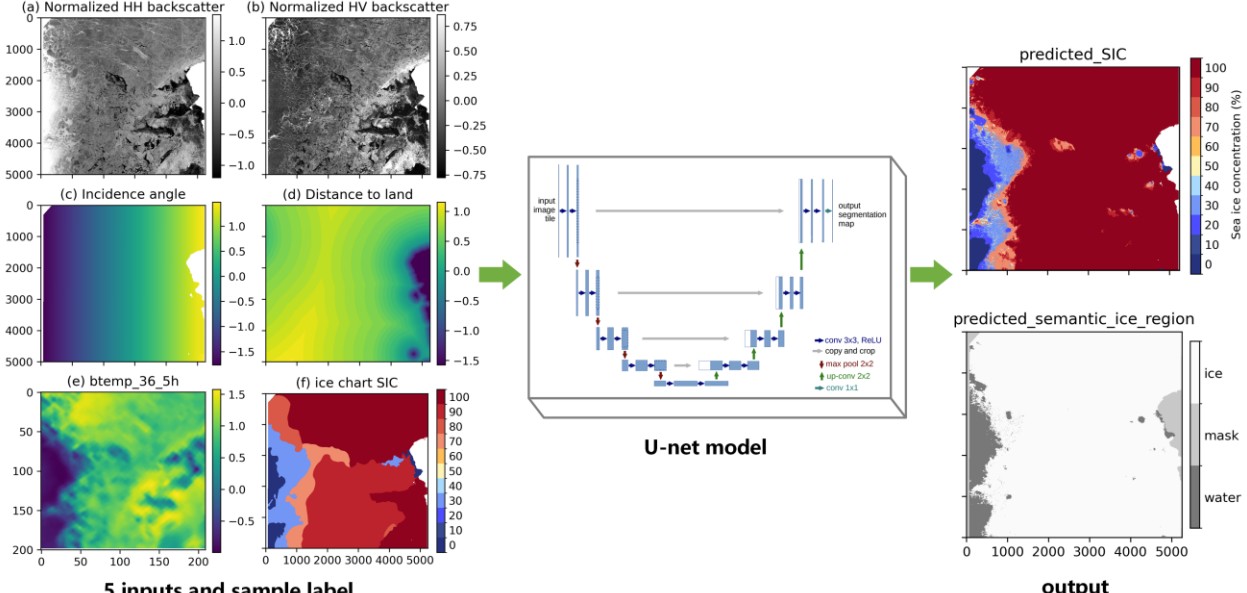

**Figure 2.** Example input and output of the simplified U-Net model. "btemp_36_5h" represents AMSR2 36.5GHz H-polarized TB data with the image size of about 200×200 pixels, which is eventually resampled to an image size of about 5000×5000 pixels consistent with
other input images. Compared with ice chart SIC at the input, the semantic segmentation of ice region predicted by U-Net model reduces the overestimated ice domain extent by about 20km and has more detailed ice edges. The sample image was captured in Fram Strait on April 29, 2021, same as that in Fig. 3 to 9.

**Table 1.** Experimental parameter setting of U-Net model

| training images | validation images | testing images | label | patch | batch | epoch | receptive field | levels | filters | SAR sigma unit | learning rate |
|---|---|---|---|---|---|---|---|---|---|---|---|
| | | | | | | | | | | | |



| 482 + repeated 19 | 30 + additive 3 | 20 | Chart SIC | $512^2$ | 16 | 80 | 764 | 6 | 16, 5×32 | dB | 1e-4 to 5e-6 |
|---|---|---|---|---|---|---|---|---|---|---|---|

We refer to three conclusions from the experiments conducted by (Kucik and Stokholm, 2023; Stokholm et al., 2023; Stokholm et al., 2022): **1)** When training with 11 SIC classes (0% to 100%, in 10% increments), the U-Net model achieved the highest accuracy for predicting 0% SIC (open water) and 100% SIC (dense pack ice); **2)** The classification-based CrossEntropy loss function performs better in accurately predicting 0% SIC and 100% SIC, whereas using a weighted loss function results in better average performance for intermediate categories (10% to 90% SIC) and lower accuracy for 0% SIC and 100% SIC; **3)** Larger receptive fields and deeper networks result in higher model prediction accuracy. Therefore, the model training used 11 SIC classes rather than just including sea ice (>0% SIC) and open water (0% SIC) classes, and used the CrossEntropy loss function with the following loss weights: [0.05, 0.05, 0.13, 0.13, 0.13, 0.09, 0.09, 0.09, 0.09, 0.075, 0.075, 0.0], where 0.0 denotes the ignored mask class weight. The original weights for the 11 SIC classes in the 512 SAR images calculated using the method from (Stokholm et al., 2022) are: [0.01, 0.55, 0.11, 0.14, 0.20, 0.12, 0.20, 0.11, 0.07, 0.06, 0.01]. These weights, approximately four levels, were adjusted to balance the prediction of 0% SIC and intermediate SIC categories, hence the weights set as previously mentioned. Considering the available memory of computer resources, a 6-layer convolutional structure with a 764×764 pixels receptive field was used to extract local texture features. However, our experiments showed that the model performs better with a patch size slightly smaller than the receptive field. Therefore, a patch size of 512×512 pixels, a batch size of 16, and 80 training epochs were used.

Additionally, the training of the U-Net model involves three experimental settings different from those used in the initial AI4Arctic U-Net model. First, the torch.optim.lr_scheduler.CosineAnnealingWarmRestarts() scheduler with an initial learning rate of 1e-4, an iteration count of 30 epochs, and a minimum learning rate of 5e-6, was employed combining with the Adam optimizer for model training. This approach helps accelerate convergence to local minima and repeatedly explores other local minima across multiple cycles.

Second, the training and testing data were reallocated to mitigate the effects of sample imbalance between different ice types or conditions. Initially, the U-Net model was trained with the following data distribution: 512 images from the AI4Arctic training dataset were randomly split into 30 validation images and 482 training images, while 20 images from the AI4Arctic test dataset were used for testing. The model was found to persistently exhibit significant errors when predicting winded open water and smooth thin ice, as these features were only a small proportion of the AI4Arctic training dataset. For instance, large areas of smooth thin ice or level first-year ice were only found in the '20190730T123155_cis_prep.nc' image in the training dataset and the '20180623T114935_cis_prep.nc' image in the test dataset. The model never accurately predicted the second image when using the first image in the training set. Therefore, the latter was also included in the training set. In the training data list, 7 images containing smooth thin ice were repeated 8 times and 12 images containing winded open water were repeated 4 times, to increase the sampling frequency of these two classes and to achieve over 99%



accuracy in open water detection. Moreover, 1 image from the former and 2 images from the latter were added to the
validation data list (see Table S1).

Third, the SAR incidence angle was introduced as a feature input, thus the patches of 512×512 pixels were directly fed
into the U-Net model without any data augmentation. The model training was conducted using a computer with two
NVIDIA GeForce RTX 3090 24GB GPUs, based on PyTorch version 2.0.1 and CUDA 11.8 modules. Training for 80
epochs required approximately 40 hours, while prediction time was around 16 seconds.

### 3.1.2 Model evaluation metrics

Model performance for 11 classes SIC estimation is assessed based on the statistical $R^2$ coefficient, as utilized and
defined in (Stokholm et al., 2022). Inspired by the MacroBins metric proposed by (Stokholm et al., 2023), we further
calculated separated recall metrics to select the best U-Net model for 2 classes ice-water segmentation.

$$OW_{recall} = \frac{water_{TP}}{water_{label}}, \quad ov40_{recall} = \frac{ice_{TP}}{ov40_{label}}, \quad bl40_{recall} = \frac{ice_{TP}}{bl40_{label}} \tag{6}$$

Here, $water_{TP}$ and $ice_{TP}$ mean the true positive pixel numbers of water (0% SIC) and ice (over-0% SIC), respectively.
$water_{label}$, $ov40_{label}$ and $bl40_{label}$ then segment three regions, that is, open water: 0% SIC, large-concentration ice region:
over-40% SIC, and low-concentration ice region: 0~40% (excluding 0%) SIC, according to the SIC classes in training labels.
In this study, we assumed the best U-Net model should detect the extents of 0% SIC and over 40% SIC labelled by ice charts,
that are definitely OW and dense pack ice, with near-100% precision. Whereas the extents with 0~40% (excluding 0%) SIC
labelled in ice charts are usually overestimated compared to SAR images. Thus, the best-performing U-Net model for
semantic ice segmentation was selected with the highest sum value of $R^2$, $OW_{recall}$ and $ov40_{recall}$ among 80 training epochs
(see Fig. 7).

### 3.2 The Multi-textRG algorithm

#### 3.2.1 Calculate GLCM textures

GLCM textures capture pixel-neighborhood correlation features within a fixed-size window, similar to the super-pixel
textures used in CRF (e.g., (Zhang et al., 2021)) and the semantic context information extracted by CNN (e.g., (Kortum et al.,
2021)). These local features are useful for distinguishing between ice types and open water with similar SAR backscatter
intensities but differing relative positions to easily recognizable ice/water pixels. The intercorrelation among GLCM texture
features and their sensitivity to different ice types have been thoroughly analyzed (Guo et al., 2023; Murashkin et al., 2018;
Park et al., 2020). To select the most effective GLCM textures, we used the code provided by NERSC (https://github.com/
nansencenter/MOIRA) and calculated 13 GLCM textures defined by (Haralick et al., 1973) in 182 Sentinel-1 HV images,
and analysed the ability of these texture features to perform ice-water segmentation using the J-M distance metric (Bruzzone





et al., 1995). Please refer to Fig. S1 in the supplement file, HV Sum Avg and HV Cont, as well as the similar HH Sum Avg and HH Cont textures were finally selected.

The parameters for GLCM texture computation were set to 64 grey levels, a window size of 32×32 pixels, a sliding step of 4 pixels, a distance of 8 pixels, and four directions: 0°, 45°, 90°, and 135°. Guo et al. (2023) noted that textures derived from SAR HH images in the logarithmic (dB) domain can be affected by the linear relationship with the incidence angle. Therefore, we computed GLCM textures based on the linear unit domain of backscatter, with the normalized range for HH polarization backscatter values set to [0, 0.2] and for HV polarization to [-0.001, 0.02] (without truncating negative values).

Limiting the selection to the Sum Avg and Cont textures under HH and HV polarization facilitates maximizing the exploration of feature combinations useful for identifying complex ice types. Figure 3 presents the HH Sum Avg and HV Sum Avg textures, which have been normalized to the [0, 64] range, as well as the logarithmic normalized HH Cont and HV Cont textures, i.e., $hhCont_n$ and $hvCont_n$. The logarithmic normalization to the [0, 1] range is achieved based on the natural logarithm within the [1, e] range:

$$scaler = \frac{(e-1.0)^{1/n}}{max(glcm)},$$

$$glcm_n = ln((glcm * scaler)^n + 1.0) \tag{7}$$

Here, $e$ denotes the natural exponential, $ln()$ represents the natural logarithm, and $max()$ refers to the maximum value function. $glcm$ denotes either the HH Cont or HV Cont texture, and $scaler$ is the scaling factor that maps the maximum texture value to $ln(e)$ (i.e., 1.0). Since the Cont texture compresses the range of low backscatter values, the exponent $n$ in the equation is set to 0.5, indicating a stretching of the lower values.

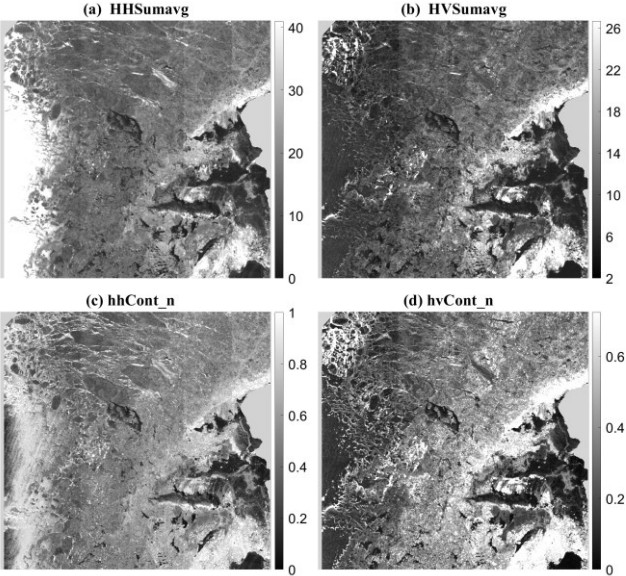

**Figure 3.** Exampled 4 textures: light grey area means land mask (Fig. 4 to Fig. 10 remain the same)



### 3.2.2 Calculate the polarization ratio

SAR dual-polarization data exhibit significantly different sensitivities to wind speeds, incidence angle, and thin ice,
making them crucial for improving sea ice classification (Boulze et al., 2020; Fors et al., 2016; Zakhvatkina et al., 2019) or estimating sea ice concentration (Karvonen, 2014). Specifically, as referenced in Fig. 8(9) and 10, the backscatter from open water in HH polarization is often significantly higher than that from sea ice at low incidence angles and is generally higher than that from open water in HV polarization under low to moderate wind speeds (Karvonen, 2014). Thin ice and melting ice surfaces have backscatter values close to zero in HV polarization, similar to calm water surfaces; however, these surfaces in
HH polarization usually exhibit non-zero backscatter, distinctly higher than that of calm open water (Cristea et al., 2022). Additionally, level first-year-ice (FYI) with smooth surfaces shows significantly higher backscatter in HV images compared to HH images (Dierking, 2013; Zakhvatkina et al., 2013). These characteristics indicate that HH and HV polarizations possess complementary image features for enhanced sea ice classification, which has been demonstrated with machine learning algorithms (Kucik and Stokholm, 2023; Wang and Li, 2020).

In this study, we introduce the HV/HH ratio and its transformed computation, defined by the following formulas:

$$inci \in [20\ 45];\ HVsumavg,\ HHsumavg \in [0\ 64],$$

$$func1(x) = 0.5 + 0.5 * tanh(x * 10 - 6),$$

$$func2(x) = 0.4 * \left( e^{-\frac{(x-0.7)^2}{2*0.3^2}} + e^{-\frac{(x-1.0)^2}{2*0.3^2}} + e^{-\frac{(x-1.3)^2}{2*0.3^2}} + e^{-\frac{(x-1.6)^2}{2*0.3^2}} \right) \tag{8}$$

$$rhvhh = \frac{HVsumavg+1}{HHsumavg} + HVsumavg/64 * func1(norm(inci)),$$

$$rhvhh_n = func2(rhvhh) \tag{9}$$

Here, $inci$ represents the incidence angle matrix. The functions $func1(x)$ and $func2(x)$ vary along the range direction as shown in Fig. 4(a). $func1(x)$, similar to a Sigmoid function, is a step function, while $func2(x)$ is an "Ω"-shaped function obtained by summing four Gaussian probability distribution functions. The range of both functions is [0, 1]. The $norm()$ function denotes proportional normalization to the [0, 1] range. In Equation (9), the addition of 1 to HVsumavg and then
dividing by HHsumavg is intended to prevent an extremely low polarization ratio caused by very low backscatter of thin ice or wetted ice (surface type 2 in Table 2) in SAR HV gray image. Figure 4 and Table 2 shows the designed outcomes of $func1(x)$ and $func2(x)$ for the calculation of HH/HV ratio and normalized HH/HV ratio. Overall, the unique advantages of $rhvhh_n$ are as follows: 1) It reliably assigns open water at low incidence angle and calm water at all incidence angles to near zero, and 2) assigns thin ice/wetted ice to significantly higher than zero. Values close to zero ensure that multiplying with
other textures results in values close to zero and dividing with other textures results in extremely high values.



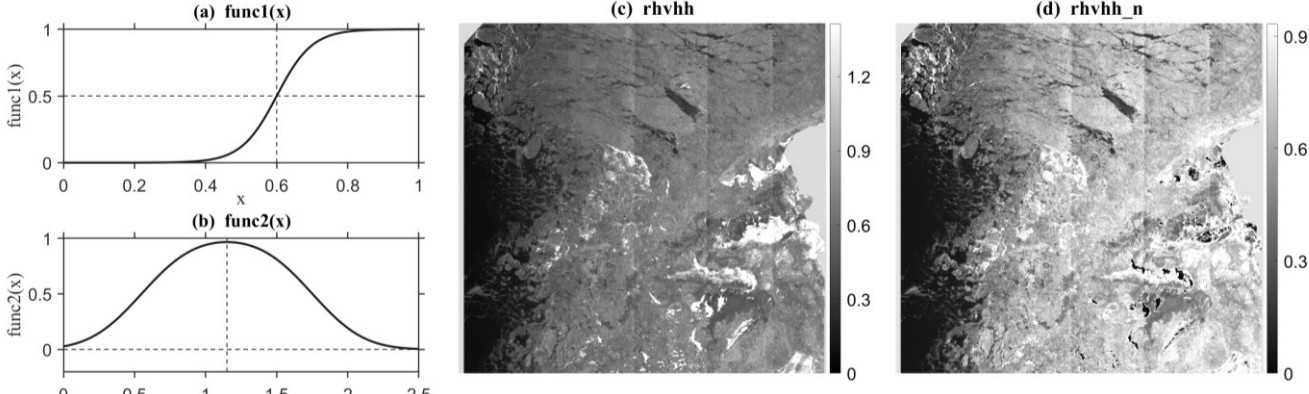

**Figure 4.** Examined HV/HH ratio feature.

**Table 2.** The recognition ability of 5 sea surface types of polar ratio feature $rhvhh_n$

| # | Target surface types | SAR backscatter values | $rhvhh$ values | $rhvhh_n$ values |
|---|---|---|---|---|
| 1 | open water at low incidence angles (to the left of Fig. 8(a) and (b)) | significantly lower than 64 in HV image, and close to 64 in HH image | close to 0 | close to 0, the darkest |
| 2 | thin ice or wetted ice at low incidence angles (top left and bottom right of Fig. 8(a) and (b)) | very close to 0 in HV image, and significantly higher than 0 in HH image | about 0.5 to 1.5 | about 1, significantly brighter than calm open water |
| 3 | winded open water at high incidence angles | lightly higher than thin ice in HV image, and higher HV grey than HH grey | about 1.5 to 2.5 | about 0 to 0.5, lower than ice surface |
| 4 | ice with medium-high brightness | higher than 0 in both HV and HH images | about 1.0 | about 1, significantly brighter than calm open water |
| 5 | calm open water (right center of Fig. 8(a) and (b)) | near 0 in both HV and HH images | significantly higher than 2.0 | near 0, the darkest |

### 3.2.3 Procedure of the Multi-textRG algorithm

Recognition of objects with low inter-class separability often requires a multi-stage algorithm, similar in concept to the flow-based cloud masking in optical data or the "one-vs-all" approach in machine learning. While neural network algorithms are prevalent in sea ice classification, the importance of automatically or semi-automatically generating reliable training labels necessitates a reconsideration of multi-stage discrimination methods. The Multi-textRG algorithm aims to leverage regional textures of images and local thresholds, integrate classic GLCM textures with simple region-growing methods, and progressively distinguish ice surfaces with different levels of backscatter intensity. The threshold separability of backscatter intensity and local textures is fundamental for SAR ice-water classification. To minimize the backscatter of winded open water or to enhance the contrast between open water and low-backscatter sea ice, we designed the following combined texture:



$$Ctext = 1 / (hvCont_n * hhCont_n * rhvhh_n) \tag{10}$$

Here, $*$ denotes scalar multiplication, the / symbol indicates scalar division, and the $\frac{1}{x}$ function significantly stretches the low value range and inverses the high-low to low-high value range. The $hvCont_n$ is advantageous for identifying ice with moderate backscatter, the $hhCont_n$ and $rhvhh_n$ are effective for detecting ice with low backscatter and winded open water, and the $hvCont_n$ and $hhCont_n$ combine the high ice-water contrast in HH or HV images. In $Ctext$, black pixels

represent ice, and white pixels represent water.

As shown in Fig. 5 and 6, the Multi-textRG algorithm uses the $Ctext$ texture and comprises three stages: 1) identifying high-backscatter ice, typically thick ice or bright ice floes; 2) identifying moderate to low-backscatter ice; and 3) identifying low/weak backscatter ice, usually thin ice or ice with melting surfaces. The algorithm accomplishes this through three times of region-growing. Every time of region growing involves selecting ice seeds, calculating grow thresholds, and expanding

new ice seeds, which are implemented with three usages of the sliding window. The detailed algorithm implementation is as follows:

**Step one: Calculate the segmentation threshold matrix $t_1$ to identify ice seeds using the "first peak principle" in 300×300 pixel sliding window with a step size of 60 pixels.** Obtain the histogram distribution of $Ctext$ values for all pixels in each window and then determine a segmentation threshold. The histograms in the top left of Fig. 5 illustrate

two scenarios for determining the threshold. First, if a low-value trough appears after the first peak in the histogram, a right triangle is constructed based on the horizontal and vertical axis coordinates of the first peak and the subsequent trough, as shown in the plot. The segmentation threshold is then determined as the intersection of the fitted histogram line and the median line of the triangle's hypotenuse, referred to as the "triangle hypotenuse median threshold method." Second, if no low-value trough is identified after the first peak in the histogram, the threshold is determined using twice

"triangle hypotenuse median threshold method" approach. In each window, the pixels with texture values less than $t_1$ are ice seeds.

**Step two: Calculate the grow threshold matrix $t_2$ in 300×300 pixel sliding window with a step size of 60 pixels.** Use the average of 5-pixel-step absolute texture difference for ice seeds to determine a grow threshold in each window and then resample the grow threshold matrix $t_2$ to the image size of the $Ctext$.

**Step three: Grow new ice seeds using $t_2$ in 61×61 pixel sliding window with a step size of 5 pixels.** In the example shown in the bottom left of Fig. 5, the yellow square indicates the current 61×61 pixel sliding window centred at pixel $(i, j)$. Within this sliding window, calculate the average texture value $a$ of all seeds, and determine the pixels with texture value $b$ satisfying $abs(b - a) < t_2$ as new ice seeds. The expanded seeds will serve as base seeds for further growing in the next sliding window.

**Step four:** Following steps one, two, and three, perform regional growing to identify high-backscatter ice pixels within the U-Net semantic ice region, as shown in Fig. 6(d) and (g), typically consisting of thick ice or bright ice floes.





**Step five:** Following steps one, two, and three, perform regional growing again to identify medium and low-backscatter ice within the residual unrecognized U-Net semantic ice region, as shown in Fig. 6(e) and (h), which includes most smooth thin ice or wetted ice.

**Step six:** Following steps one, two, and three, perform regional growing thirdly to identify the lowest backscatter ice within the residual unrecognized U-Net semantic ice region, as shown in Fig. 6(f) and (i), which includes small area of smooth thin ice or wetted ice.

The Multi-textRG algorithm runs on a computer with an Intel(R) Core(TM) i5-10400F CPU @ 2.90GHz and 64GB RAM, taking approximately 270 seconds for SAR image denoising, around 580 seconds for GLCM texture calculation, and

about 540 seconds for three-stage ice detection using the Multi-textRG algorithm.

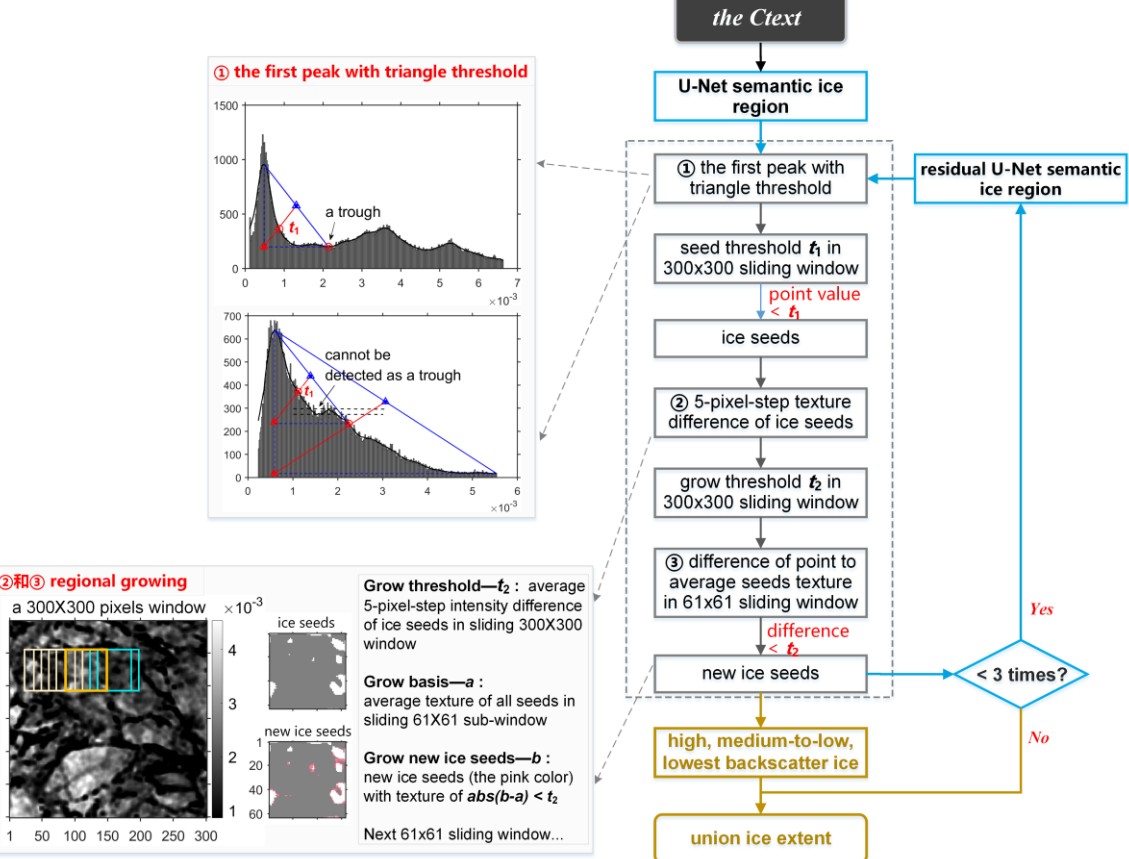

**Figure 5.** The procedure of the Multi-textRG algorithm.





**Figure 6.** The exampled 3-stage ice detection of Multi-textRG algorithm: (a) is the combined texture **$Ctext$**; (b) is the U-Net semantic ice region (see Fig. 2), from which the sea ice edges are extracted and marked as the brown contour in (d)-(i). (c) The final ice detection result generated by the "logical or" operations of (g), (h) and (i). The visual growth of ice pixels mainly occurs near the ice edges.





## 4 Results

### 4.1 Prediction result of U-Net model

Figure 7(a) shows the training accuracy curves of the U-Net model, including training loss, validation loss, $R^2$, and three recall values. Due to the iteration cycle of learning rate scheduler as 30 epochs, both the training loss and validation loss curves exhibit a periodic fluctuation trend. In the first two cycles, the loss values show a declining trend, while in the third cycle, the loss values fluctuating higher because of the model begins to exhibit some overfitting. The $R^2$ and three recall values show similar periodic changes, with $bl40_{recall}$ curve exhibiting the most fluctuations. This indicates that the U-Net prediction difficulty majorly lies in identifying regions with 0%-40% SIC. The sum of $R^2$, $OW_{recall}$, and $ov40_{recall}$ reaches its highest at the 56th epoch, indicating the best-performed U-Net model.

Figure 7(b),(c) and Figure 7(d),(e) depict two examples including the ice chart SIC and the U-Net predicted SIC maps. Despite the experiment setting making efforts to reduce sample imbalance among different ice conditions by setting class weights in the optimizer and increasing image proportions of special ice conditions, the U-Net model still struggles with recognizing intermediate SIC categories. The significant imbalance in 11 SIC classes and the coarse labels in ice charts may have posed considerable challenges in predicting intermediate SIC categories. Developing a more accurate SIC prediction CNN model requires training on more balanced samples and finer-grained sample labels.

Despite that, the U-Net model performs well in segmenting the semantic ice region while using a 30% SIC threshold, see Fig. 7(c) and Fig. 7(e) (comparing to Fig. 8(9) and Fig. 10). As presented in Table 3, the $OW_{recall}$ and $ov40_{recall}$ values in validation and test scenes are both above 97%, though the $bl40_{recall}$ is only around 70%. Results show that when $OW_{recall}$ and $ov40_{recall}$ are close to 100%, the U-Net model achieves accurate ice region detection regardless of $bl40_{recall}$. In 0%-40% SIC region, the large inaccuracy of ice charts is caused by the coarse grids ranging from several kilometers to tens of kilometers, whereas the 40 m resolution of SAR image allows for further separation of ice and water pixels. In over-40% SIC region, differently, dense pack ice appears at SAR image resolution, not allowing further separation of ice and water pixels.

Many studies indicate that the primary challenges in SAR-based sea ice classification lie in wind-disturbed open water and low backscatter thin ice (including newly formed ice, level young ice, and level first-year-ice et.al.) or wetted ice surfaces (covered by wet snow or melt ponds). Thin ice and wet ice surfaces typically appear in dense ice regions with over-40% SIC. Conversely, brighter and more easily recognizable ice surfaces, including fractured ice floes, brash ice, and newly formed frazil ice, usually appear in the 0%-40% SIC regions. The $OW_{recall}$ and $ov40_{recall}$ in Table 3 exceed 97%, indicating that the U-Net excels in identifying winded open water and low backscatter thin ice or wet ice. Thus, the sequential Multi-textRG algorithm can precisely detect ice pixels unaffected by wind-disturbed water surfaces.



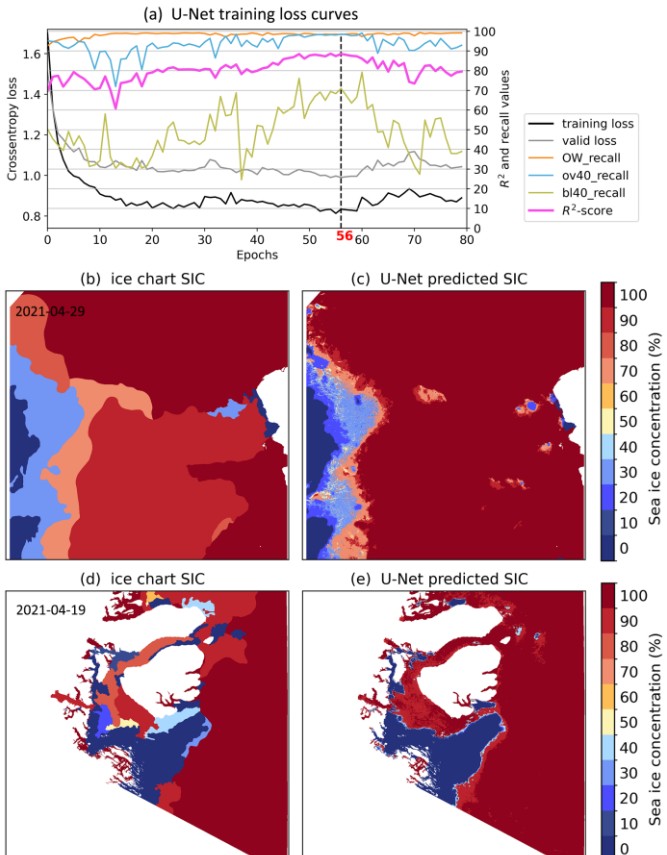

**Figure 7.** Training accuracy curves and prediction map of the U-Net CNN model. The regions with over(including) 30% SIC is segmented as the major ice regions.

**Table 3.** Validation and test accuracy of U-Net CNN model.

| Based on 33 validation scenes | | | | Based on 20 testing scenes | | | |
|---|---|---|---|---|---|---|---|
| validation accuracy: $R^2$ | $OW_{recall}$ | $ov40_{recall}$ | $bl40_{recall}$ | testing accuracy: $R^2$ | $OW_{recall}$ | $ov40_{recall}$ | $bl40_{recall}$ |
| 88.503% | 98.337% | 98.547% | 70.979% | 87.066% | 97.343% | 98.993% | 79.789% |

## 4.2 Ice detection result of Multi-textRG algorithm

### 4.2.1 Example 1: Thin ice

Combining the U-Net model and Multi-textRG algorithm, the primarily SAR-based ice-water classification is achieved. Five examples with various ice conditions are shown in Fig. S2 to show the classification results from U-Net model and Multi-textRG algorithm. Here, Figure 8 and 9 show the validation of SAR detected ice extent from the Multi-textRG



algorithm against sea-ice signs from Landsat-8 and Sentinel-2 optical data. The cloud-masked mosaic data from Landsat-8
and Sentinel-2, exported from the GEE platform, are presented in blocks (red grids). Both optical datasets overlap in the
upper right of the SAR image, where thick ice with high backscatter dominates. Additionally, Sentinel-2 data covers a large
area of thin ice with low backscatter in the lower right of the SAR image. Validation metrics in Fig. 8 include an OA of
96.9%, an FN of 1.02%, an FP of 2.06%, and a $P_{area}$ of 9.7%. In Fig. 9, the validation metrics include an OA of 73.0%, a
relatively high FN of 4.73%, a significantly higher FP of 22.3%, and a $P_{area}$ of 16.9%.

Firstly, in the upper right of Fig. 8, there are some black point and black line features in the SAR HV gray image, while
the corresponding area in the SAR HH gray image does not display dark features. The Landsat-8 and Sentinel-2 natural color
images show white or gray in Fig. 8(d) and 9(d), indicating nearly 100% dense pack ice in this region. Thus, the black point
and line features observed in SAR HV gray image may represent darker thin ice in ice leads (referred to as dark ice leads).
This difference between SAR HH and HV gray images reveals a critical physical mechanism of surface scattering: rough
surfaces can lead to a greater proportion of incoherent reflected waves, meaning the polarization of reflected waves tend to
differ from that of incident waves, resulting in stronger HV-polarized backscatter (Beaven et al., 1994). Conversely, smooth
surfaces are predominantly governed by coherent scattering, with a weaker depolarization effect, indicating that the
polarizations of incident and reflected waves tend to be similar, resulting in stronger HH-polarized backscatter (Lievens et al.,
2022). In the Multi-textRG algorithm, the dark ice leads are accurately identified by using the $rhvhh_n$ feature.

Secondly, in the lower right of Fig. 9, the black regions in SAR HH and HV gray images correspond to smooth thin ice
surfaces covered with dry snow. The image was taken on April 29, 2021, at the Fram Strait (80°N to 82°N), when and where
dry snow remains. In optical images, the interaction mechanism of wavelengths shorter than 3 μm with surfaces is primarily
reflective (König et al., 2001). The snow albedo in the visible spectrum is influenced mainly by impurities, grain size, and
liquid water content (König et al., 2001). The blue or light gray surfaces in Fig. 9(d) represent wet snow or wet ice, while the
completely white area in the lower right part of Fig. 9(d) indicates dry snow. C-band microwaves can penetrate dry or
refrozen snow to depths ranging from several meters to tens of meters (Rott and Nagler, 1993), allowing the retrieval of
backscatter from the snow-ice interface. However, when the snow thickness is substantial, volume scattering may occur
within the snow due to density anisotropy caused by impurities, potentially enhancing the signal's depolarization effect
(Lievens et al., 2022). The low backscatter in SAR HV gray image indicating weak depolarization, suggests that the area
consists of thin and dry snow, which allows the microwave to pass through and thus observe the smooth thin ice surface
beneath.

These large areas of smooth thin ice closely resemble open water in both contour and texture characteristics, making
them difficult to distinguish using raw GLCM textures and SAR dual-polarization data. Under high incidence angles
(approximately 38°-46°), the smooth snow-ice interface acts as a near-specular scattering surface, resulting in low
backscatter values in both HH and HV polarizations. The smooth thin ice and thin, dry snow both tend to increase coherent
scattering while reducing incoherent scattering. When zooming into local regions in Fig. 9(f) and 9(g), the backscatter from
smooth thin ice in HH polarization is not absolutely zero, showing irregular, intersecting, faint linear features, whereas the



backscatter in HV polarization is closer to zero, appearing smooth and nearly featureless surface. The Multi-textRG algorithm utilizes $\frac{1}{rhvhh_n}$ to significantly enhance the low backscatter surfaces in HH image, thus the regional growing

method successfully identifies most smooth thin ice.

However, due to the limitations of C-band SAR observations, some smooth thin ice may appear as the darkest in both HH and HV polarizations. In such case, $rhvhh_n$ cannot effectively differentiate it from calm open water. As illustrated in Fig. 9(f), the smooth ice surface within the blue contours is erroneously classified as open water. These thin ice surfaces, even in the Sentinel-2 natural color image in Fig. 9(g), appear smoother than the surrounding ice, indicating a more mirror-

like reflective characteristic and extremely low backscatter.

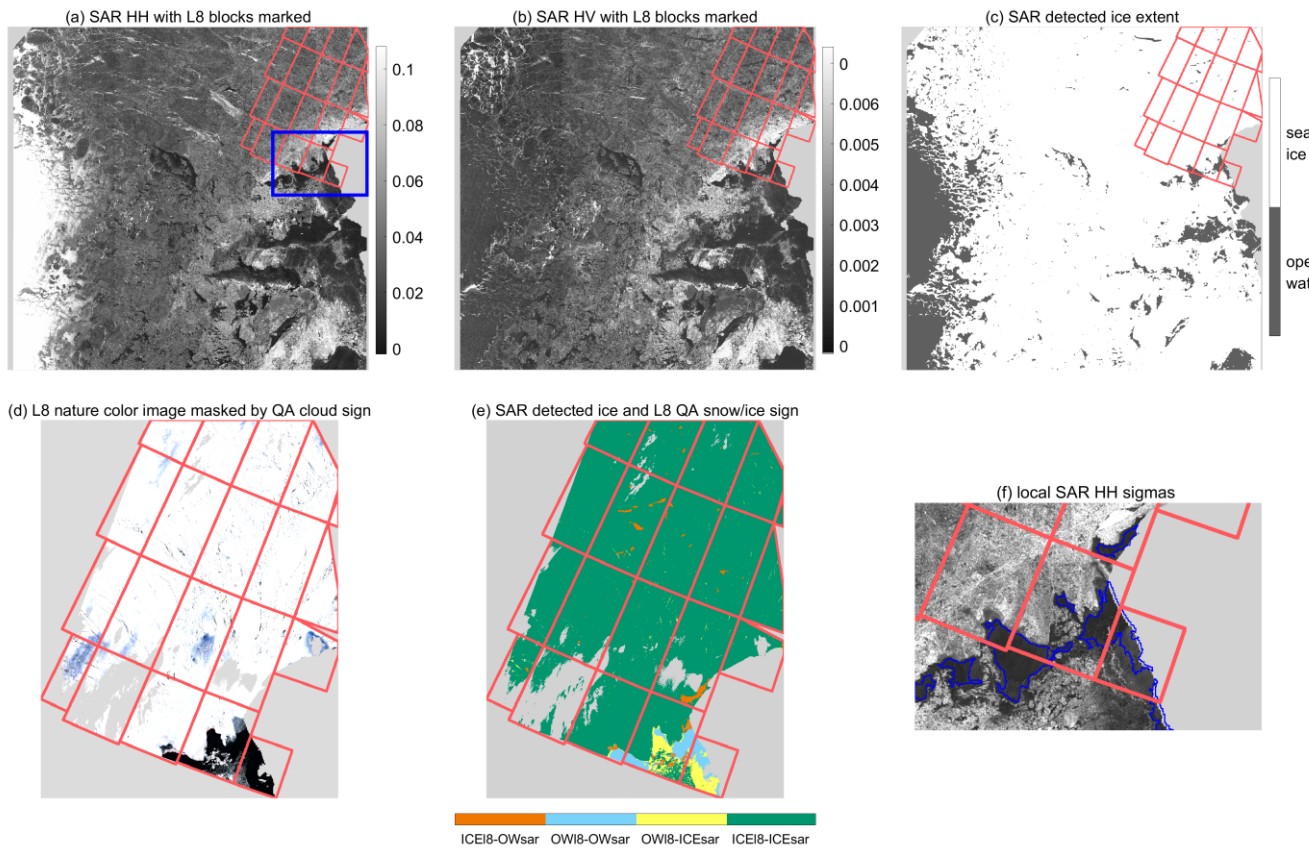

**Figure 8.** Example one: thin ice, the validation based on the Landsat 8 (L8) data for SAR detected ice extent: (a) and (b) are SAR HH and HV polarization backscattering grayscale images, (c) is the SAR detected ice extent obtained by the Multi-textRG algorithm, (d) is L8 RGB band true color image, (e) shows the comparison between SAR detected ice extent and L8 QA snow/ice sign. (d) and (e) are views

narrowed down to the effective values region of L8, (f) zooms in on the local SAR HH image within the blue box region. The red grids show the position of the optical image, the Nan value is masked with light gray, and the blue curves are SAR detected ice edges, which are all same with that in Fig. 9 and 10.



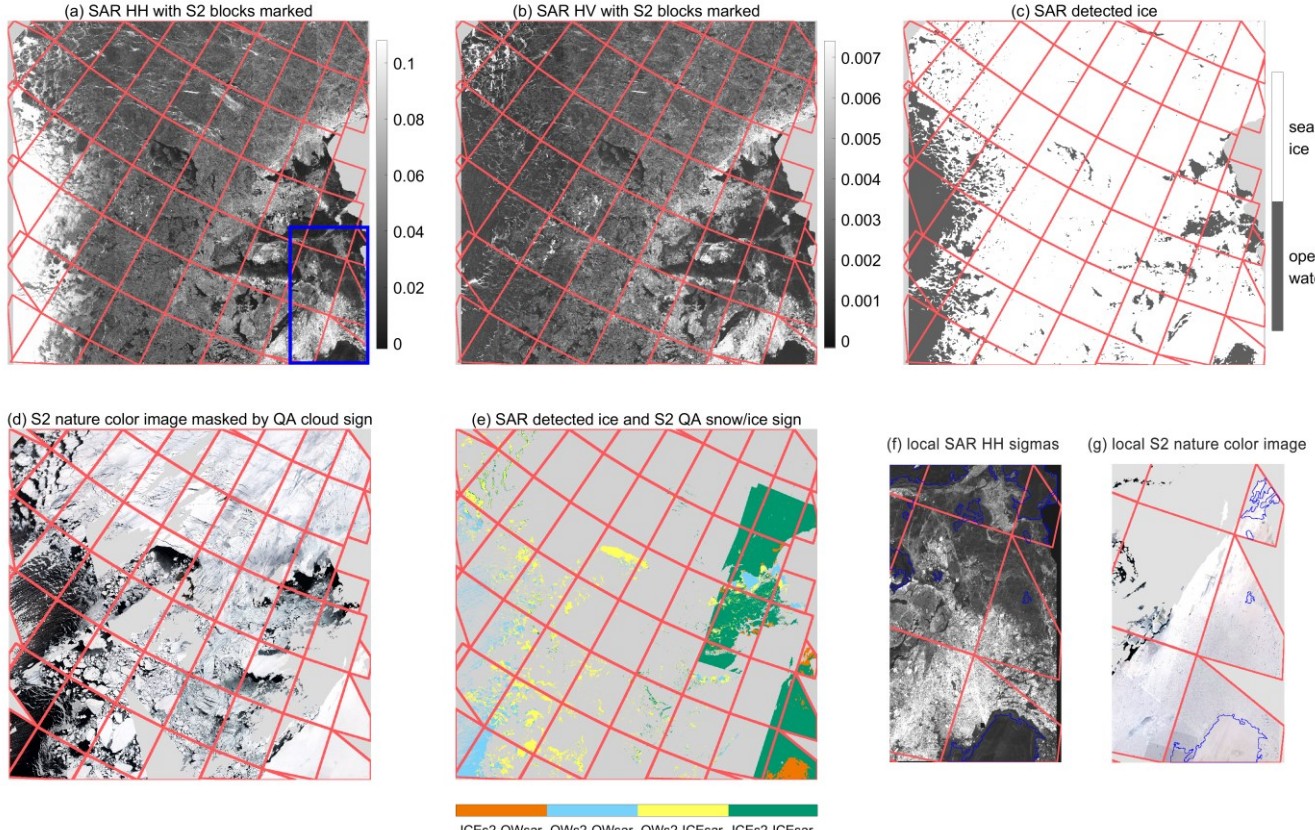

**Figure 9.** Example one: thin ice, the validation based on the Sentinel-2 (S2) data for SAR detected ice extent: (a) to (c) are same with Fig.8, (d) is S2 RGB band true color image, (e) shows the comparison between SAR detected ice extent and S2 QA snow/ice sign. (f) zooms in on the local SAR HH image within the blue box region and (g) zooms in on the local S2 nature color image.



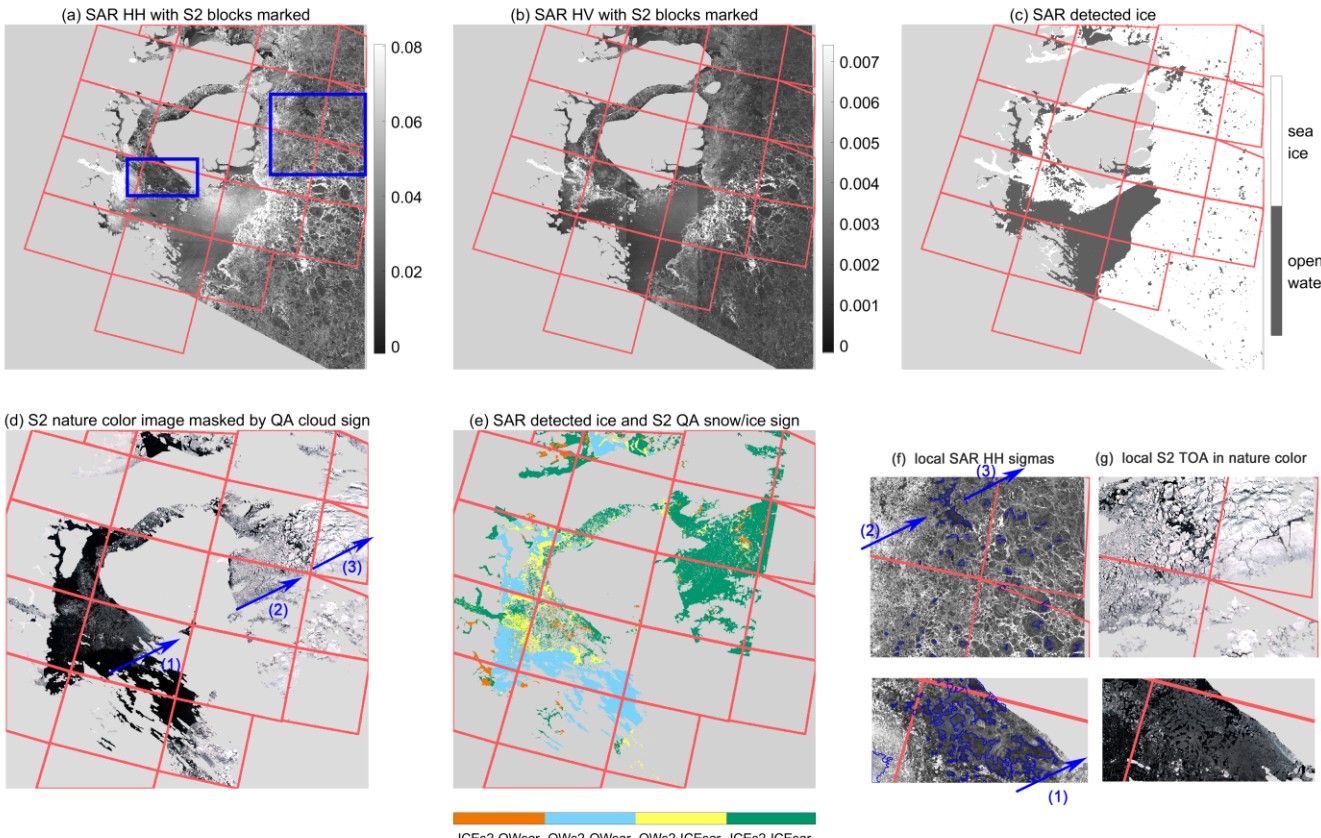

**Figure 10.** Example two: wetted ice and winded open water, the validation based on the Sentinel-2 (S2) data for SAR detected ice extent: (a) to (e) are same with Fig. 9. (f) zooms in on the local SAR HH image within the blue box region and (g) zooms in on the local S2 nature color image. The three arrows (1), (2) and (3) indicate the possible offshore wind direction. Therefore, the ice melt degree and snow cover condition of the three stages are different, the ice albedo and backscattering coefficient are different.

### 4.2.2 Example 2: wetted ice, winded open water

Figure 10 shows the SAR detected ice result for another image, validated against Sentinel-2 optical data. In (a) and (b), the low backscatter ice within and around the blue rectangular box and the bright winded water surface at the image center are clearly distinguishable. (c) demonstrates that the U-Net and Multi-textRG algorithms successfully identified winded open water, as well as the low backscatter circular melt ponds or snow-covered ice regions. Meltwater reduces both optical albedo and radar backscatter. In (d), along the segments and directions indicated by the three arrows, the ice surface albedo gradually increases, suggesting a decrease in meltwater on the ice/snow. In SAR HH and HV gray images, the ice surface at the first arrow (within the small rectangle) appears nearly black, while at the second and third arrows (within the large rectangle), the ice surface exhibits similar brighter backscatters and texture features. This suggests the possible presence of strong offshore winds along the arrow direction, promoting ice melting upstream of the wind and causing snowfall or existing snow to accumulate downstream, resulting in differences in ice surface albedo. Radar microwaves can penetrate the



snow and return from the snow-ice interface, thus resulting in similar backscatter values in the second and third arrow segments.

In Fig. 10(f) and (g), the dark water gaps clearly visible in the high-resolution Sentinel-2 images, however are not discernible in the coarser resolution SAR HH and HV gray images and are instead identified as dense pack ice. In the second and third arrow segments, there are many low-backscatter melt ponds similar black to those observed in the first arrow segment. However, the high optical albedo suggests they may be refrozen meltwater accumulated on various-sized floes, forming new smooth thin ice covered by snow and surrounded by rough ice edges or ridges formed by floe deformation. The
validation accuracy for Fig. 10 includes an OA of 83.9%, a FN of 5.38%, a FP of 10.7%, and a $P_{area}$ of 11.4%.

Based on the dual-polarization scattering or reflection characteristics of C-band microwaves on thin ice surfaces analyzed in Example 1, the backscatter value of refrozen melt ponds in the HH polarization is also slightly higher than in the HV polarization. In Example 2, the Multi-textRG algorithm successfully identified most refrozen melt ponds but failed to fully recognize ice surfaces with meltwater or wet snow because of their extremely low backscatter in C-band SAR dual-
polarization observations. Extracting sufficient features to distinguish between extremely low backscatter wetted ice and calm water requires larger texture computation windows. For instance, the width of the black ice areas within the rectangle in Fig. 9 and the small rectangle in Fig. 10, can reach up to 1000×1000 pixels in SAR image with 40 m spatial resolution, making it difficult for the GLCM textures calculated in 32×32 pixels window to capture enough contextual information to distinguish it from calm water. Again, utilizing larger contextual local features is typically advantageous for CNN models.

Moreover, the winded open water in Fig. 8 and Fig. 10 were accurately identified using SAR HH/HV polarization ratio feature. The winded open water appears on the left of Fig. 8(b), characterized by distinct wind streaks, and in the scattered bright areas at the center of Fig. 10(a) and 10(b). The surface backscatters are significantly higher than those of thin ice and wetted ice, making it challenging to distinguish between the three types using thresholds of HH or HV backscatters. In this study, the U-Net CNN model was trained based on dual-polarization SAR images, accurately segmenting the ice surface and
winded open water using semantic contextual features from large convolutional fields. Subsequently, the high sensitivity of HH image to winded water at near-to-medium incidence angles is conversed to near-zero values in $rhvhh_n$, the Multi-textRG algorithm thus distinguishes thin ice and wetted ice from winded open water.

However, as shown in Fig. 10(f) and (g), the spatial resolution differences between optical and SAR images along with the resolution reduction from window-based GLCM texture computation, lead to some overestimated SAR sea ice extent.
They cause the high FP values of 24.3% and 10.3% in Fig. 9 and Fig. 10, respectively. The first reason is visually discernible, and the second can be explained as follows: at 40 m pixel resolution, the sharp gradient at ice edges causes the HV Cont texture within at least one 32×32 pixel GLCM window to be high, thus misclassifying water as ice in at least one window. Distance of the detected ice edge from the actual ice edge depends on the window size and sliding distance parameters during GLCM texture computation. Fig. 9(f) and 10(f) illustrate a noticeable distance between the blue contour lines and the
white ice edges. Comparing the yellow ($OW_{s2} - ICE_{sar}$) regions in Fig. 9(d), 9(e) and Fig. 10(d), 10(e), small ice floes were masked as clouds, leading the Multi-textRG algorithm to overestimate the SAR ice extent along these floe edges.





## 4.3 Validation accuracies

In this study, we combined the U-Net model and the Multi-textRG algorithm to achieve detailed ice identification for 530
SAR images provided by the AI4Arctic project dataset. Figure 11 presents the scatter accuracy metrics for the comparison
and validation of ice detection results of 219 images against Landsat-8 and Sentinel-2 optical data. The results show that the
area-weighted average OA is 84.9%, with an average FN of 4.24% and an average FP of 10.8%. As analyzed in Section 4.2,
the lower FN value indicates a small underestimation for low backscatter ice surfaces (thin ice or wet ice), highlighting the
advantage of the Multi-textRG algorithm. In contrast, the higher FP value reflects a larger overestimation along the ice edges,
stemming from the pixel resolution differences between SAR ice extent (160 m) and visible optical data (30 m), with a
maximum discrepancy not exceeding a few kilometers.

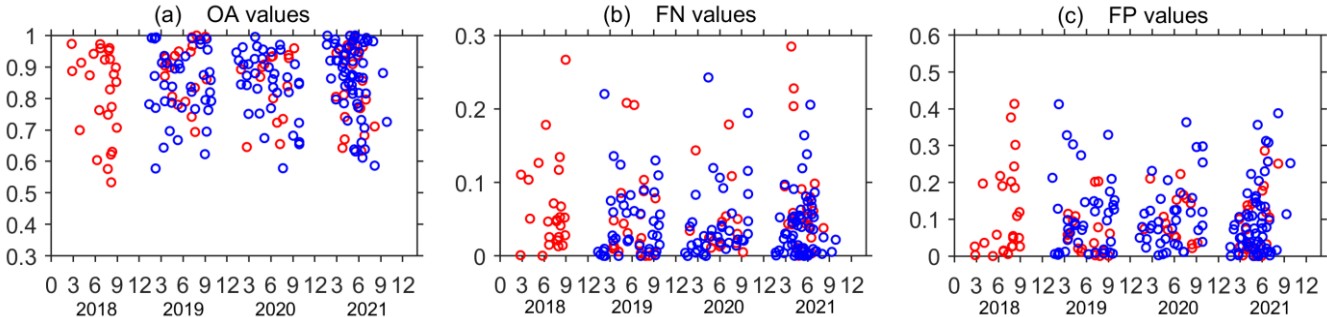

**Figure 11.** Validation based on optical data for U-Net+Multi-textRG produced ice extent: (a), (b) and (c) show the scatter values of the
OA, FN and FP evaluation metrics, where the red and blue circles represent Landsat-8 and Sentinel-2 as validation data, respectively.

## 5 Discussions

### 5.1 Selection and processing for GLCM textures

Figure S1 shows that Sum Avg and Cont have significant advantages in distinguishing complex ice surfaces. The former
represents the average backscatter (normalized to 0~64 gray levels) and the latter represents the contrast sensitive to edge
features, within the sliding window. In this study, the Sum Avg and Cont texture calculations of dual-polarization data were
used to compute the polarization ratio $rhvhh_n$ and the combined texture $Ctext$, maximizing the contrast and separability
between ice and water. Different GLCM textures may have their advantages in identifying different sea ice surfaces. For
example, the Cont texture has a wide histogram width, compressing the texture values of low backscatter surfaces, while the
IDM (or Corr) texture has a narrow histogram width, compressing the texture values of high backscatter surfaces (see HV_1
and HV_4 texture maps in Fig. S1). In preliminary experiments, both Cont and IDM were used to design combined texture
features, however it was found that they are highly correlated. The compression of texture values for low or high backscatter
surfaces can be achieved through certain functional transformations, such as inverse functions ($1/x$) or logarithmic functions
($log(x)$). Therefore, the choice of Cont texture can be replaced by other GLCM texture features.



## 5.2 Performance of the U-Net and Multi-textRG algorithms

The validation result demonstrates that the proposed U-Net and Multi-textRG algorithm framework effectively addresses the challenge of accurate ice-water classification under complex ice conditions, with the overall accuracy reaching 84.9%.
However, the ice leads or broken gaps with widths of tens of meters, clearly visible in 30 m resolution optical data, are almost completely unidentifiable in 40 m resolution SAR data. The backscatter values and texture features of fractured ice and dense pack ice surfaces show few differences in SAR images, leading to a higher FP value (10.8%) in the validation errors. As well, temporal differences between optical and SAR data acquisition also result in significant ice drift in some images, contributing to parts of the FP and FN (4.24%) errors.

Another source of the higher FP value comes from the presence of both winded open water and fragmented ice. As shown in Fig. S3, when winds blow over the fragmented ice area and waves pass through it, the ice-wave interaction may cause wave breaking and result in densely high backscatter values. A large proportion of ice floe edges become indistinguishable. There is no better way to differentiate between fragmented ice and wind-disturbed water surfaces based on SAR backscatters and textures, thus perhaps excluding such images from usage is a better choice.

The FN value also comes from low-lying ice surfaces where meltwater accumulates, such as a corner of a melt pond, as shown in Fig. 10. The Multi-textRG algorithm maximizes the identification of melt ponds, remaining small unrecognized proportion of melt ponds because of the nearly zero backscatter values in both C-band SAR HH and HV polarizations. The L-band, with its greater penetration depth and lower sensitivity to melt ice (Singha et al., 2018), might address this issue when different radar band SAR observation data are fused.

Further experiments in larger polar regions are necessary to test the robustness of the algorithm framework. In the proposed U-Net combined with Multi-textRG algorithm framework, the U-Net CNN model is replaceable. This algorithm framework can serve as a good reference for fine ice type classification or other classification studies.

## 6 Conclusions

We propose a novel algorithm framework combining a supervised U-Net CNN model and an unsupervised Multi-textRG
algorithm, using Sentinel-1 SAR and GCOM-W1 AMSR2 data as input, to achieve fine-grained ice-water classification, and then validate the classification results with Landsat-8 and Sentinel-2 optical data.

Firstly, provided by the AI4Arctic project, the U-Net model uses SAR dual-polarization (HH, HV) GRD data and AMSR2 36.5GHz H-polarized data as input, SIC in DMI/CIS ice charts as sample labels, and the identification accuracy of open water with 0% SIC and dense pack ice with over-40% SIC as evaluation metric, to segment semantic ice region in
overall. During the training of the U-Net model, the experimental parameters were set based on several main conclusions of existing studies. It was found that the number balancing of training samples under various ice conditions in SAR images is crucial for the model prediction performance. By increasing the sampling rate of level first-year ice (or level young ice) and





winded open water, the U-Net model effectively distinguishes between ice surfaces and winded water surfaces, achieving nearly 100% identification accuracy for open water and dense pack ice areas.

Secondly, within the U-Net segmented ice region, the Multi-textRG algorithm combines the logarithmically normalized HH Cont and HV Cont GLCM textures (i.e., $hvCont_n$ and $hhCont_n$) and the two-sides normalized HV/HH polarization ratio (i.e., $rhvhh_n$), uses the inverse product of three textures as classification basis, and then conducts three times of region growing to identify ice pixels with high, medium-to-low, and extremely low backscatters in SAR images. The multi-staged and texture-relied algorithm has two main advantages: 1) The combined texture feature, with that $hvCont_n$ and $hhCont_n$ serve for identifying the majority of middle-to-high backscatter ice and $rhvhh_n$ fully utilize the incidence angle effect of HH polarization data to discriminate winded open water and low backscatter ice, highlights the complementary effect of dual-polarization data. 2) The region growing calculates the seed threshold and growing threshold in slide windows fully exploiting the regional gradual similarity and local sharp contrast over various ice surfaces in SAR images. Compared with completely unsupervised image segmentation methods, the Multi-textRG algorithm allows for some manual parameter adjustments (e.g., the thresholds and the window size), enabling fast and accurate sea-ice labeling.

There have always been significant challenges in SAR sea ice classification algorithms for identifying certain ice types, including winded open water, smooth thin ice, new ice, and melting ice surfaces. The proposed algorithm framework most likely addresses these challenges. A CNN model utilizes semantic context information to differentiate complex ice surfaces and open water combining SAR and AMSR2 data input. The Multi-textRG algorithm then achieves multi-stage ice detection in SAR images similar to multi-stage cloud identification in optical images, without being disturbed by winded water surfaces. Validating the SAR classification results with Landsat-8 and Sentinel-2 optical data, an overall OA value of 84.9% was acquired. An overall average FP value of 4.24% indicates a small underestimation of low backscatter ice surfaces. An FN value of 10.8% means a relatively larger ice overestimation along ice edges due to the resolution difference between SAR and optical data. Future work can combine an advanced CNN model with the Multi-textRG algorithm to improve the method robustness in larger Arctic oceans. The designed HH and HV polarization ratio and the combined GLCM texture may also contribute to develop possible SIC estimation CNN methods.

**Code and data availability**

Codes, testing data, and the visual validations of our ice-water classification results to Sentinel-1 and Landsat-8 optical data are available at https://zenodo.org/records/13269639.

**Author contributions**

The methodology, experiments, analysis and manuscript writing were implemented and carried out by Yan Sun. Experiment result analysis, experiment redesign, validation and manuscript content organization were developed in collaboration with



Shaoyin Wang, Teng Li, Chong Liu, Yufang Ye, Xi Zhao. The study was carried out under the supervision and support of Xiao Cheng. All authors have reviewed the manuscript.

**Competing interests**

The authors declare that they have no conflict of interest.

**Acknowledgements**

This study was supported by the National Natural Science Foundation of China (Grant Number: 41925027, 42206249), and the Innovation Group Project of Southern Marine Science and Engineering Guangdong Laboratory (Zhuhai) (Grant Number: 660 311023008). The authors would like to thank the AI4Arctic team who provided the ready-to-train dataset and the well-designed U-Net model (https://platform.ai4eo.eu/auto-ice/data). We also greatly thank the European Space Agency (ESA) for providing Sentinel-1 and Sentinel-2 satellite data and the United States Geological Survey (USGS) for providing Landsat-8 satellite data. The insightful comments from the reviewers and editors are highly acknowledged.

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
