# Peer review of "Combining the U-Net model and a Multi-textRG algorithm for fine SAR ice-water classification"

_EGUsphere, 2024_

## Author Comment (AC1)

**Response to the referee 1**

Thank the referee for taking time to review our manuscript. We really appreciate your time and your review.

Note: The (Page x, Line xx-xx) gives where the explanation or revision locates based on the preprint of our manuscript egusphere-2024-2760.

The revision based on the referee's comments will be uploaded together with the reply to all the reviewers later.

In this manuscript a U-Net model combined with Multi-textRG algorithm has been proposed for fine ice-water classification on SAR imagery from AI4Arctic dataset. Overall the manuscript is clearly written with sufficient details. However, as U-Net and other techniques such as GLCM feature extraction has already been broadly used for sea ice mapping and particularly in AI4Arctic dataset. The manuscript should further emphasize the novelty in terms of methodology in this research, as well as comparison with previous methods as baselines. Detailed comments are listed below.

- **Comment 1:** The authors mentioned in the abstract that the proposed algorithm successfully addresses ice–water classification across all seasons. However, during the evolution of sea ice, the proportion of ice types presented by different seasonal patterns is unstable. In warmer seasons, melting ice surfaces affect the classification results, while in colder seasons, snow cover on sea ice also influences the outcomes. The authors did not evaluate the algorithm's performance under varying environmental conditions, indicating a lack of demonstrated adaptability and effectiveness in different seasonal contexts. This limitation should be clearly emphasized and analyzed.

  **Response 1:** Yes, we agree with the reviewer's comments about the evolution of sea ice. As for our algorithm's performance under varying environmental conditions, we can explain it from two respects:

  **1)** The SAR scenes provided in the AI4Arctic dataset were selected by the experts from the Technical University of Denmark (DTU), Danish Meteorological Institute (DMI), and Nansen Environmental and Remote Sensing Center (NERSC), considering ice/water type diversity and season balance **(Stokholm et al. 2022)**. In this paper, we used the AI4Arctic dataset in the latest Version 2 released on May 25, 2023. **Figure 1** (a) and (b) in our manuscript respectively show the spatial location and seasonal distribution of the SAR images, i.e., the 512 SAR images have

a relative balanced distribution across all seasons with the maximum number in July to September and the minimum number in December to February. Considering the complicate ice conditions within the waters around Greenland and the Canadian Arctic Archipelago, thus, we said the AI4Arctic dataset "features typical classification challenges of ice and water surface characteristics while maintaining a certain feature balance" **(Page 4, Line 105-106)**. please also see the analysis in Section 2.5 **(Page 7, Line 205-213)**.

2) Our algorithm figured out the ice-water classification of these 512 SAR images in high accuracy and high resolution, as the examples shown in **Figure 9, Figure 10, and Figure S2** in the supplement PDF file. Besides, we have made the ice-water classification results of 219 (85+134) SAR images with visual validations to Sentinel-1 and Landsat-8 optical data available at https://zenodo.org/records/13269639, so that the reviewer and other readers can look through the visual result performance.

Except for the focus on method statement, we have added two sentences in the Conclusion section to emphasize the typicality of SAR images provided by the AI4Arctic dataset and the accessible link of our more validation results. The sentences are:

"The 512 Sentinel-1 SAR images provided by the AI4Arctic dataset include abundant ice conditions. The proposed algorithm developed based on this dataset likely addresses the accurate ice-water classification across all seasons."

These words are added before the "Future work …" at **(Page 27, Line 644)**.

- **Comment 2:** The authors stated that the framework is primarily designed for ice-covered regions and wind-driven open water areas. However, wind forcing can also affect the classification accuracy within ice-covered regions. Therefore, the authors should clarify how wind-driven dynamics influence the classification performance across different types of ice-covered areas.

**Response 2:** We resolved the identification on most of the wind-driven open water areas from the contiguous thick ice (means large brightness) and densely smooth ice, except for a certain condition, such as the broken/fragmented thin ice and wind-driven open water mixed region. We have given out the explanation in the Discussion section 5.2, referring to **Figure S3** in the supplement PDF file.

- **Comment 3:** In the third paragraph, the authors claimed that the algorithm combining CNN with empirical methods represents the optimal automatic approach

for sea ice labeling. However, the evidence supporting this conclusion appears overly strong, as there is insufficient experimental validation to substantiate the claim of optimal performance. Multiple experimental results are needed to support such a conclusion. Furthermore, the proposed algorithm lacks comparative analysis with other sea ice classification methods, whether quantitative or qualitative, which further weakens the assertion of its superiority.

**Response 3:** Yes, comparative analysis with certain other algorithms is essential. However, due to space limitations, this paper primarily focuses on detailing our methodology and presenting intuitive results, as well as the validations. We acknowledge that our initial phrasing lacked sufficient rigor—without extensive comparisons with other methods, the "optimal" conclusion should be expressed more cautiously.

When referring to our approach as "optimal" in the introduction, we base this claim on theoretical justification of its methodological advantages. That is, the combination of supervised CNN in coarse semantic segmentation and unsupervised empirical method in detailed pixel classification most likely achieve the fine ice-water classification **using coarse labels**, comparing to the collected methods in the third paragraph. This is the subjective conclusion of the review of current research status in the Introduction. We did not claim that our algorithm is the optimal in the Conclusion section.

But now, the technological advancements continually redefine the state-of-the-art. For instance, recent work in **(Zhang et al. 2025)** addresses the same image classification challenges in high-resolution remote sensing images, i.e., **"**Ultra High Resolution (UHR) remote sensing imagery (RSI) (e.g. $10{,}000 \times 10{,}000$ pixels) poses a significant challenge for current Remote Sensing Vision Language Models (RSVLMs). If choose to resize the UHR image to standard input image size, the extensive spatial and contextual information that UHR images contain will be neglected. Otherwise, the original size of these images often exceeds the token limits of standard RSVLMs, making it difficult to process the entire image and capture long-range dependencies to answer the query based on the abundant visual context.**"**, consistent with our review discussion in the fourth paragraph **(Page 27, Line 84-88)**.

Zhang et al. (2025) achieves a balance between high-resolution detail and large-context learning through an advanced Remote Sensing Multimodal Large Language Model (RSMLLM), i.e., the ImageRAG model, enabling purely deep learning-based high-precision image classification. In contrast, our method

combines deep learning with traditional classification method. However, the ImageRAG model uses the manually annotated MME-RealWorld dataset as benchmark, whereas our algorithm framework just uses the coarsely labelled ice charts, which significantly enhances its practicality for operational applications.

- **Comment 4:** Line 105, the related publication concerning the AutoIce Challenge should be mentioned here (doi: 10.5194/tc-18-3471-2024), which would facilitate readers to refer to this particular challenge and its details.

  **Response 4:** Thank you for your suggestion, we will add this reference (Stokholm et al. 2024) and (Stokholm et al. 2022) in the revised manuscript.

- **Comment 5:** Line 120, the U-Net-based model has already been further improved by the AutoIce participants using a bunch of techniques and achieved relatively high accuracy in the AI4Arctic dataset (illustrated in doi: 10.5194/tc-18-3471-2024 and doi: 10.5194/tc-18-1621-2024). According to Fig. 2, it seems that the U-Net used in this research has the same architecture as the one used in the challenge. Therefore, it is necessary to illustrate how the U-Net-based method proposed in this research different from the previous ones. It is also necessary to implement those U-Net-based models as benchmarks to compare with the proposed method in the manuscript.

  **Response 5:** Yes, we employed the same U-Net architecture provided by the AI4Arctic competition. However, our goal is the ice-water classification. Based on coarse SIC labels in ice charts, the original U-Net model yields SIC classification results with rough ice boundaries as depicted in **Figure 7**. Our algorithm further achieves refined identification of ice pixels, as illustrated in **Figure 9**. **Figure S2** presents five additional example images for reference.

  Although the U-Net model remains unchanged, the data preprocessing methodology applied in this study was optimized through extensive experimental validation. Notably, the input SAR images have significant thermal noise, to which the U-Net performance is highly sensitive, as demonstrated in previous research **(Stokholm et al. 2022)**. In the initial manuscript submitted to the TC AI4Arctic special issue, we utilized the denoised and normalized Sentinel-1 SAR data provided by the AI4Arctic competition. However, the classification results still have a higher false-negative (FN) ratio for wetted ice surfaces. In this manuscript, we also conducted substantial sensitivity investigations but not presented and we reprocessed raw Sentinel-1 SAR images by implementing an alternative denoising algorithm combining that proposed by (Sun and Li 2021) and by (Park et al. 2020).

Consequently, the key distinctions between our algorithm and the original U-Net model are: 1) different image preprocessing methods leading to different feature inputs; and 2) superior spatial resolution of the output in our results.

**References**

[1]  Park, J.-W., Korosov, A.A., Babiker, M., Won, J.-S., Hansen, M.W., & Kim, H.-C. (2020). Classification of sea ice types in Sentinel-1 synthetic aperture radar images. The Cryosphere, 14, 2629-2645

[2]  Stokholm, A., Buus-Hinkler, J., Wulf, T., Korosov, A., Saldo, R., Pedersen, L.T., Arthurs, D., Dragan, I., Modica, I., Pedro, J., Debien, A., Chen, X., Patel, M., Cantu, F.J.P., Turnes, J.N., Park, J., Xu, L., Scott, K.A., Clausi, D.A., Fang, Y., Jiang, M., Taleghanidoozdoozan, S., Brubacher, N.C., Soleymani, A., Gousseau, Z., Smaczny, M., Kowalski, P., Komorowski, J., Rijlaarsdam, D., van Rijn, J.N., Jakobsen, J., Rogers, M.S.J., Hughes, N., Zagon, T., Solberg, R., Longépé, N., & Kreiner, M.B. (2024). The AutoICE Challenge. The Cryosphere, 18, 3471-3494

[3]  Stokholm, A., Wulf, T., Kucik, A., Saldo, R., Buus-Hinkler, J., & Hvidegaard, S.M. (2022). AI4SeaIce: Toward Solving Ambiguous SAR Textures in Convolutional Neural Networks for Automatic Sea Ice Concentration Charting. Ieee Transactions On Geoscience and Remote Sensing, 60, 1-13

[4]  Sun, Y., & Li, X.-M. (2021). Denoising Sentinel-1 Extra-Wide Mode Cross-Polarization Images Over Sea Ice. Ieee Transactions On Geoscience and Remote Sensing, 59, 2116-2131

[5]  Zhang, Z., Shen, H., Zhao, T., Guan, Z., Chen, B., Wang, Y., Jia, X., Cai, Y., Shang, Y., & Yin, J. (2025). Enhancing Ultra High Resolution Remote Sensing Imagery Analysis with ImageRAG. IEEE Geoscience and Remote Sensing Magazine, 14

---

## Author Comment (AC5)

**Response to the editor**

**Thank the editor for giving us this valuable opportunity to finalize our manuscript. We really appreciate your kind comments.**

**Major revisions of the manuscript we have made are added at the latter part of this manuscript. Please see the bookmarks on the left side in the PDF file for quick and clear guidance.**

We further reviewed the comments from two reviewers and the editor, i.e., CC1, CC2 and EC1. We would like to conclude them as the following **five** major points.

**Point 1:** To further explain the motivation and necessities to integrate the U-Net with the Multi-textRG algorithm.

**Revision 1:** We reorganized the structure of **Section Introduction**. From the importance of precise sea ice classification, to SAR and PMW data fusion advantages, to the algorithm status of SAR-based ice classification, to the algorithm challenges, to the supervised deep learning relying on limited fine-grained sample labels, to more available ice charts labels, to combination of region-based U-Net and pixel-based Multi-textRG, and to their single effects.

We have made extensive revisions to the Introduction with a clearer restructured logic. These improvements reflect the insights gained from our most important cognition that have settled after all this time, as well as the most meaningful perspective guided by the comments from the reviewers and editors. We particularly appreciate your contributions for enhancing the quality of this manuscript.

**Point 2:** To state the innovation of this article is Multi-textRG algorithm and introduce clearly about its advantages.

**Revision 2:** We have always stated that the U-Net model is provided by the AI4Arctic. But, we did new training by ourselves with new Sentinel-1 SAR image processing. While the data processing changed and the prediction goals changes (SIC estimation changed to binary SIE segmentation), the model training most likely changes as well. Actually, it does.

In the last paragraph of Section Introduction, we have emphasized the respective function of U-Net and Multi-textRG. We believe we have indicated that the Multi-textRG is the major newly proposed one. "Moreover, Multi-textRG can operate independently when rough SIE references (such as NSIDC IMS data) are available. However, considering operational applicability and the reliability of third-party data, this paper remains to elaborate on the training configuration of the Sentinel-1/AMSR2-based U-Net model and details the critical data preprocessing steps."

**Point 3:** To introduce the diversity of ice conditions contained in the AI4Arctic dataset.

**Revision 3:** We included several sentences to explain the ice condition diversity in SAR images in the AI4Arctic dataset. They are "The Sentinel-1 SAR images in the AI4Arctic dataset cover the waters surrounding Greenland and the Canadian Arctic Archipelago from January 8, 2018, to December 21, 2021. These regions are characterized by rapid ice drift, continuous and fast-changing freeze or melt processes throughout the year, high wind speeds over open water, and frequent offshore wind streaks, all of which pose significant challenges for SAR-based sea ice classification algorithms. Certain

challenging types, even for professional ice analysts, are difficult to accurately interpret during ice chart production (Stokholm et al., 2022). As shown in Fig. 1, the SAR images exhibit uniform temporal and spatial distribution patterns. Therefore, the AI4Arctic dataset is highly valuable**"**, locating at the original Section 2.5 (data coverage).

The description was put long after the Section 2.1 (AI4Arctic project dataset). The skipped presentation may introduce extra reading difficulty. Therefore, we have revised to move Figure 1 into the beginning of Section 2 (data). Besides, we have added related words in the next-to-last paragraph in the Section 1 (Introduction) to clarify the advantage of AI4Arctic dataset. Please check that.

**Point 4:** To reduce the text describing the U-Net algorithm and results, and conduct a comparison between the U-Net and Multi-textRG results.

**Revision 4:** We have appropriately streamlined the description of the U-Net algorithm and results, i.e. the Section 3.1 (The U-Net model experimental settings) and the Section 4.1 (Prediction result of U-Net model).

**There is no doubt about the improvement of ice edge details in Multi-textRG results compared to U-Net results. Because the coarse ice labels for U-Net training defined the region-level ice detection output.** If the Multi-textRG had no improvement compared to U-Net results, we would not have written this paper and would not have made all the codes public. In the original manuscript, **we used the Figure.S2 in the supplement PDF file to show their difference.** We think this is enough. Many journal articles, after all, use exampled images to state the algorithm effectiveness. But, we understand that the claim "we anticipate that the proposed algorithm framework successfully addresses accurate ice-water classification across all seasons" needs full-scale demonstrations. We achieved high quality ice-water classification for almost all the SAR images in AI4Arctic dataset, which shows high data diversity and data reliability. Thus we used the word "anticipate" to state the accuracy of the expression. We **revised the related sentence in the Abstract as** "Through detailed analyses and discussions of classification results under the highly diverse ice and water conditions included in the AI4Arctic dataset, we anticipate that the proposed algorithm framework successfully addresses accurate ice-water classification across all seasons and enhances the labelling process for ice pixel samples.".

Moreover, we have finished the FP and FN metrics calculation for our U-Net outputs and the Landsat-8/Sentinel-2 optical QA ice/snow signs. The comparison between U-net predicted SIE and optical validate data results in an average OA of **83.7%,** false negative (FN) of 2.22%, and false positive (FP) of 14.3%. Meanwhile, the validated metrics for the Multi-textRG detailed SIE include an average OA of **84.9%**, false negative (FN) of 4.24%, and false positive (FP) of 10.8%. Their statistical scatter maps are respectively shown as below.

[Figure]

Validation metrics for U-Net outputs.

[Figure]

Validation metrics for Multi-textRG results (Fig. 11 in the original manuscript).

**Fig. R1.** Validation of the comparison between U-net predicted SIE (with a 20% SIC threshold) and optical (Landsat-8/Sentinel-2) data.

As shown in Fig. R1, the two group of metrics show small differences. This is because the filtered optical images mostly include densely packed ice, i.e. near-100% sea-ice covered regions. Despite that, some individual validation images including broken ice MIZ display effective detail and precision improvement of Multi-textRG results to U-Net outputs. Fig. R2 further show the comparison of several examples imaged in MIZ. Considering the small metric differences, as well as earlier words we have discussed, we decide to keep removing the comparison between U-net predicted SIE and optical ice/snow detections. But we have updated to upload all the visual comparison results of U-net predicted SIE and optical ice/snow detections at https://zenodo.org/records/13269639. We also include Fig. R2 into the supplement PDF file in the revised version.

[Figure]

**Fig. R2.** Examples with broken ice in MIZ to state the comparison between the U-net outputs and the Multi-textRG results.

**Point 5:** The accuracy of Landsat-8/Sentinel-2 QA ice/snow sign remains to be proven. The Multi-textRG results are not adequately compared with the results of other classical algorithms.

**Revision 5:** We do not have good idea to prove the accuracy of Landsat-8/Sentinel-2 QA ice/snow sign at present. But we have made open access to all Landsat-8/Sentinel-2 optical images, and we include **Fig. R2** in the revised supplement file.

The Multi-textRG results are not adequately compared, for this question, we would like first to explain the reasons for so detailed analyses about the validation of Multi-textRG results to Landsat-8/Sentinel-2 optical images under thin ice, wetted ice and winded open water conditions. It is because we found the three ice/water conditions were the persistent source of identification errors, as we have reviewed in the Section 1 (Introduction). The two examples of Fig. 8 to Fig. 10 are precisely intended to emphasize that Multi-textRG algorithm finally achieved their accurate classification with high robustness and high quality for ice edge description. The two examples represent the most outstanding performance of this algorithm.

What's more, this outstanding performance was further demonstrated within the added validation to a pure-CNN SAR SIE product (Wang and Li 2020). **See Page 9 to 11 in this PDF file.**

**The Editor Comments and our one-to-one replies are below:**

After assessing the manuscript, the reviews, and the authors' replies, I have noted that much of the concerns about the manuscript may arise from some confusion about the intent of the combined technique of the U-Net and Multi-textRG, and the degree of novelty and/or validation for each. I urge the authors to carefully address the reviewer concerns in their manuscript, and I also add some comments below that I believe, based on the reviews, are key points to address:

- **Editor Comment 1:** One reviewer has noted that the U-Net is essentially the same as previously published, without any substantial improvement in results. The authors' contend the use is as a first step in classification. As such, the paper is primarily about the Multi-textRG pixel level classification scheme. I would encourage the authors to make this more clear and downplay the U-Net analysis. To me, this appears as just a technique to first provide regional classifications. **In fact, I believe it may not be a necessary step-one could use a variety of techniques to do a first classification.** The real innovation in the paper is the Multi-textRG algorithm, and the paper could be more clear on this.

  **Response 1:** Yes, this paper is primarily about the Multi-textRG algorithm and the U-Net model is not a necessary step-one. Other semantic segmentation models can also achieve effective coarsely ice region detection. The AI4Arctic SAR dataset includes various ice surface features. During the U-Net model training, preprocessing of AI4Arctic SAR dataset and the configuration of compensation for ice type imbalance are of great importance. Therefore, in Section 3 "Methods", we have appropriately streamlined the description of training metrics. In Section 4.1 "Prediction result of U-Net model", we condensed the content to ensure conciseness.

Moreover, we revised the Section 1 Introduction to make it more clear that the Multi-textRG algorithm is our major contribution.

- **Editor Comment 2:** Following from this, it is not clear to me, based on my understanding of the Multi-textRG algorithm, that the first step regional classification is even necessary at all. Since the pixel level classification can identify ice, open water, or wind-affected open water within the regional ice classes, then can it not do the same for areas that were outside of these original classes? If I misunderstood this, then the authors should clarify why the first step is needed. If I am correct, I don't think the authors need to remove the U-Net classification, but it should be clarified why it may be advantageous to use (or a similar technique). Why the regional segmentation was needed is not clear.

  **Response 2:** Thank the editor for raising this point. The step-one of U-Net semantic segmentation is necessary. The region-growing method consists of two steps: 1) selecting initial ice seeds based on GLCM texture intensities, and 2) growing new ice seeds based on their texture intensity difference from previous ones. During the first step, it is critical to ensure that all selected ice seeds are true ice pixels. To achieve this, we restrict seed selection to within ice regions segmented by the U-Net. Without this step, due to the similar or higher intensity values of wind-roughened open water compared to smooth thin ice, some water pixels may still be misidentified as ice seeds—even when the designed combined texture $C_{text}$ have enhanced ice–water contrast. If false ice seeds are selected, water pixels may erroneously grow from them, leading to incorrect ice–water classification. We have revised to emphasis on the necessity of U-Net semantic segmentation in the last paragraph of Section 1(Introduction).

- **Editor Comment 3:** The reviewers make a valid point about how well the Multi-textRG algorithm works vs state-of-the art. I appreciate the performance metrics that are included, but the reader is left to judge for themselves whether the results are actually acceptable. Even though in some cases presented, the results appear subjectively good, there is no context presented for how good these actually are compared to alternatives. In some cases, the false positive and false negative rates might be viewed as unacceptable. This is the critical factor if the manuscript is to be acceptable for publication in The Cryosphere. Based on my assessment of the reviews, I feel the authors need to show that the Multi-textRG algorithm works effectively compared to standard classification schemes, or other approaches previously proposed.

  **Response 3:** The effectiveness of Multi-textRG algorithm are actually guaranteed by both U-Net prediction accuracy and itself. There is independent evaluation metrics for U-Net results, i.e., $OW_{recall}$ and $ov40_{recall}$, of which over-97% values mean the U-Net segmentation is highly close to the ice charts. Ice charts are accurate, thus the U-Net results are accurate, and thus the Multi-textRG algorithm have no bigger errors. Because the Multi-textRG algorithm, in fact, just like to remove some extra water pixels from the U-Net segmented ice region.

  How to compare with other standard classification schemes? Apart from deep learning algorithms, many other algorithms are difficult to replicate. When we first have the idea to combine

CNN network semantic segmentation with an ice pixel detection method, we searched much to downloaded the codes within our reviewed papers. However, without one methods can be easily accessed (like the IRGS methods (Jiang et al. 2022; Leigh et al. 2014)) or repeated. Moreover, sea ice classification methods based on SAR images are usually rather complicated. Thus, in such a short time, we are very sorry that we do not have enough energy to reproduce the excellent algorithms of other papers.

Considering the editor's and reviewers' insistence, we decided to utilize a Sentinel-1 SAR SIE product produced by using the integrated U-Net, a pure-CNN SIE product (accessible at https://www.scidb.cn/en/detail?dataSetId=771301999089025024&version=V3). We have revised Section 4.3 to be the title of "Multi-textRG results validated to pure-CNN SAR SIE". Concise analyses are given here. Data introduction in Section 2, validation in Section 4 and conclusion in Section 6 have added related discriptions.

- **Editor Comment 4:** The reviewers express some concern about the quality of the training data for the Multi-textRG algorithm. I agree that there is a question about how well ice classes are classified in that imagery. It is challenging to show definitively that their classification of these images is robust, but some additional evaluation and discussion of this is warranted to give the reader a sense of how much to trust the results.

**Response 4:** In this manuscript, we focus on the precise ice and water classification, rather than the ice types/classes classification. We focus on proposing a novel method, of which the robustness is currently manifested by the diversity of sea ice types contained in the training SAR data within AI4Arctic dataset. Personally, I have processed tens of thousands of Sentinel-1 SAR images, thus we highly approve of the reliability and effectiveness of this dataset. The real robustness of the algorithm framework of a CNN model and the Multi-textRG algorithm is far from being proven. Application in larger data volume and evaluation in higher quality are still needed.

With the reviews feedback, it seems that the comparison of the limited optical images after cloud masking did not sufficiently reflect the accuracy of our ice-water classification results. Therefore, we have included more optical comparison images in the revised supplement PDF file. Moreover, we have added a comparison with the publicly available ice-water classification data , as it has been responded earlier.

**The major revision of the manuscript with the track change:**

**1) The Introduction section**

Arctic sea ice acts as a critical component in the global climate system by reflecting solar radiation, moderating heat and moisture exchange between the ocean and atmosphere, and influencing global ocean circulation patterns. The changes in sea ice extent (SIE) is closely monitored and analysed by various researchers or centers around the world. High-precision and high-resolution remote sensing

monitoring of polar sea ice has consistently been a core technological focus in the field of modern climate science.

Passive Microwave **(PMW)** radiometers and Synthetic Aperture Radar **(SAR)** sensors are the essential tools for monitoring SIE. SAR offers high spatial resolution and all-weather observation, yet constrained by incomplete daily spatial coverage and reduced capability in discriminating thin ice/wetted ice from wind-roughened open water (Sun et al. 2023). In contrast, PMW radiometers operate at kilometer-scale resolutions and are susceptible to clouds/fogs, meanwhile providing complete daily spatial coverage and superior performance in detecting thin ice/wetted ice (Sun et al. 2023). The data fusion of two sensors can combine their respective strengths in sea ice observation, enabling high-precision operational monitoring of sea ice.

SAR-based (including SAR/PMW-fused) ice classifications **face challenges of** various scales and confusing ice surface features among different ice types/development periods. Smooth thin ice (including newly formed ice, level young ice, and level first-year ice) and wetted ice (such as melt ponds and wet snow covered ice), along with calm open water, generally exhibit extremely low backscatter values (Cristea et al. 2022; Niehaus et al. 2023; Song et al. 2021; Zakhvatkina et al. 2017). Three types  typically also have similar regional semantic context and texture features  except for the linear features of ice leads . Bright young ice and heavily deformed ice (HDefI), as well as dark young ice and deformed ice (DefI), show highly overlapping SAR backscatter and GLCM texture histogram distributions (Guo et al. 2023). Newly formed ice in disk-shaped melt ponds  has extremely low backscatter  and  high-contrast boundaries with surrounding ice, particularly in Sentinel-1 SAR HV images, leading to largely potential misclassification as open water (Sun et al. 2023). Additionally, winded open water could have significantly higher backscatter and texture values than smooth thin ice, even reaching the levels of thick multi-year ice (Boulze et al. 2020; Li et al. 2021; Song et al. 2021). Various surface characteristics (such as different degrees of deformation, melting, geometric roughness, or internal micro-roughness, salinity, and wind speed) and radar parameters (such as wavelength, polarization, and incidence angle) are crucial factors influencing SAR backscatter echoes (Guo et al. 2023; Lohse et al. 2020; Song et al. 2021). Thus, previous methods have tried to combine image clustering segmentation with supervised machine learning (the second group of methods) or use the supervised CNN networks (the furth group of methods) to integrate image context and pixel information for high-resolution ice classifications.

The precondition is that, SAR pixel-level sea ice classification based on **deep learning requires finely annotated training samples.** Given the diverse ice surface characteristics, a deep learning model also demands to learn from thousands of samples. But currently, no such sea ice dataset ensures both precision and balanced feature representation. Numerous advanced vision models have been designed for general zero-shot classification tasks, such as SAM (Kirillov et al. 2023), GDINO (Liu et al. 2024), and CLIP (Radford et al. 2021), and specifically designed for zero-shot semantic segmentation of remote sensing images, such as GeoRSCLIP (Zhang et al. 2024b) and Text2Seg (Zhang et al. 2024a). However, they are primarily designed for small-object segmentation. For objects like sea ice, which exhibit large-scale continuous spatial coverage yet complex and varying surface characteristics, a single SAR image (e.g., 10,000×10,000 pixels) may represent only one typical ice condition needing to be recognized based on the entire scenario. This is comparable to a single clipped patch (usually smaller than 512×512 pixels) in urban building classification tasks. As a result, computational limitations and mismatched objectives make it difficult to apply general-purpose

advanced vision models to sea ice classification tasks. Therefore, previous approaches have relied on tens of manually annotated SAR images and simple CNN networks for training. The limited diversity of ice conditions leads to predictable low model robustness. However, from another perspective, leveraging the current accumulation of international sea ice monitoring research, there is an abundance of coarse annotations available—such as ice charts from Arctic nations, IMS data from NSIDC, and passive microwave-based SIC products. We aim to address ice-water classification across all seasons under various marine conditions, which necessitates large-scale and diverse ice chart annotations.

**The AI4Arctic Challenge** has also recognized the importance of fusing SAR and PMW data, the strong performance of deep learning in extracting image features, and the usability of large-volume ice chart annotations. Accordingly, it provides a well-preprocessed dataset including Sentinel-1 SAR, GCOM-W1 AMSR2, and ice charts along with a pre-trained U-Net model, with the aim of promoting international research on automated ice monitoring algorithms (Stokholm et al. 2024). The SAR images in the AI4Arctic Dataset were carefully selected by a multi-disciplinary team of experts (Stokholm et al. 2024; Stokholm et al. 2022), featuring typical classification challenges of ice and water surface characteristics while maintaining a certain feature balance. The highly diverse SAR data, full-channel AMSR2 data and highly reliable ice chart annotations can be well visualized. We believe the robustness of this dataset provides support for the development of highly reliable algorithms.

With these insights, **the goal of this study** is to explore fine SAR ice-water classification using coarse but abundant annotated data provided by the AI4Arctic dataset. Usually, deep learning algorithms can partially refine the details of recognition results based on coarse annotations. However, even with SAR data input at tens of meters resolution, it remains challenging for these algorithms to produce predictions at the native SAR resolution when annotations are only roughly delineated at kilometer scales. Probably, only a traditional empirical segmentation method can address the issue. **This paper proposes** to combine the pre-trained U-Net semantic segmentation network with an empirical statistical method termed the Multi-textRG pixel-level segmentation algorithm. **The U-net is essential** to reduce certain SIE overestimation in the Marginal Ice Zone (MIZ) based on coarse ice charts, and to leverage semantic context information within a large receptive field for accurate segmentation of winded open water and wetted/thin ice. **The Multi-textRG** means a multi-layer GLCM texture-based regional growing method. It has the ability to perform unsupervised pixel-level sea ice recognition within the semantically segmented ice region, i.e., without considering winded open water disturbance. Moreover, Multi-textRG can operate independently when rough SIE references (such as NSIDC IMS data) are available. However, considering operational applicability and the reliability of third-party data, this paper remains to elaborate on the training configuration of the Sentinel-1/AMSR2-based U-Net model and details the critical data preprocessing steps. Finally, we used respectively Landsat-8/Sentinel-2 optical QA band snow/ice signs and a purely U-Net predicted SAR-based ice/water classification product (Wang and Li 2020) to validate the accuracy of Multi-textRG outputs.

**2) Beginning of the Data section**

In this paper, we used 532 Sentinel-1 SAR images provided in the AI4Arctic dataset as the benchmark. GCOM-W1 AMSR2 36.5GHz channel data and ice charts provided in the same AI4Arctic dataset, Landsat-8 and Sentinel-2 Level-2 optical satellites downloaded from the Google Earth Engine (GEE) cloud collections were used and individually resampled to Sentinel-1 SAR scenes. Besides, corresponding Sentinel-1 SAR-based U-Net predicted ice/water classification images covering 2019 to 2021 (Wang and Li 2020) was used are the second validation data. Fig. 1 shows their spatial and

temporal distributions.

The Sentinel-1 SAR images in the AI4Arctic dataset cover the waters surrounding Greenland and the Canadian Arctic Archipelago from January 8, 2018, to December 21, 2021. These regions are characterized by rapid ice drift, continuous and fast-changing freeze or melt processes throughout the year, high wind speeds over open water, and frequent offshore wind streaks, all of which pose significant challenges for SAR-based sea ice classification algorithms. Certain challenging types, even for professional ice analysts, are difficult to accurately interpret during ice chart production (Stokholm et al. 2022). As shown in Fig. 1, the SAR images exhibit uniform temporal and spatial distribution patterns. Therefore, the AI4Arctic dataset is highly valuable. In Fig. 1(a), the northernmost point imaged by Landsat-8 and Sentinel-2 optical satellites reaches the northern coast of Greenland. However, the number of usable scenes is significantly limited due to widespread cloud contamination. As shown in Fig. 1(c) and (d), a total of 85 SAR images matched with valid Landsat-8 data after cloud masking; a total of 134 SAR images matched with valid Sentinel-2 data (launched in February 2019) after cloud masking; of them, 54 SAR images had matching data from both Landsat-8 and Sentinel-2. The optical images from the two satellites are densely distributed during the melt season and early freeze-up season. Data information and processing will be further introduced below.

**3) Validation to classical CNN produced SIE product of the Results section**

**4.3 Multi-textRG results validated to pure-CNN SAR SIE**

Figure 11 shows the Sentinel-1 SAR HV images, our U-Net predicted SIC maps ( ≥30% SIC is used to segment  SIE), the Multi-textRG resulted SIE, and the pure-CNN SAR SIE (Wang and Li 2020). Comparison between the U-Net output (using the 30% SIC contour) and the Multi-textRG result reveals that the latter exhibits greater clarity in capturing detailed outer ice edges and inner open-water boundaries. This visually proves the precision improvement achieved by the Multi-textRG approach. While comparison between the Multi-textRG results and to the pure-CNN SAR SIE demonstrate that the former show more complete identification of (a) broken ice, (b) brash ice, (c)(d) thin ice, and (e) level first-year ice with low backscatter intensities or subtle texture features. Additionally, the Multi-textRG algorithm provides more accurate detection of wind-roughened open water under low incidence angles. It overcomes the accuracy limitation of pure-CNN models on ice-water classification with incidence angle effect and enhances the texture intensity of flatten FYI (see Section 3.2.1 and 3.2.2) by leveraging our newly designed combined texture feature $C_{text}$. On the other hand, pure-CNN models often struggle to simultaneously capture high-resolution details for pixel-level identification and large-scale semantic contexts for region-level detection (Zhang et al. 2025), particularly when trained on imperfect labels in complex ice conditions (Li et al. 2021). This may further explain the relatively inferior performance of the pure-CNN approach.

Figure 12

 presents the  accuracy metrics for the comparison  of Multi-textRG results of 219 images against the Landsat-8 -Sentinel-2 optical ice/snow signs and 261 images against the Sentinel-1 SAR pure-CNN SIE data (Wang and Li 2020). Figure 12(a) to 12(c) show that the area-weighted average OA is 84.9%, with an average FN of 4.24% and an average FP of 10.8%, whereas the same metrics for Fig. 12(d) to 12(f) are 91.1%, 4.77% and 4.12%. Firstly, optical ice/snow signs are regarded as the "TRUE" ice detections with the highest spatial resolution. Thus, As

analyzed in Section 4.2, he lower OA consistent with the higher FP of optical validation indicates  a larger overestimation along  ice edges stemming from the pixel resolution differences between  Multi-textRG SAR SIE (160 m) and Landsat-8/Sentinel-2 ice data (30 m). While the close FNs of both optical and SAR pure-CNN validation indicate a small underestimation for low backscatter ice surfaces (thin ice or wet ice) in Multi-textRG results. However, the overall average metrics just show the basic consistency of different SIE observations. The superior ice recognition capability of the Multi-textRG algorithm compared to the pure-CNN model in complex ice conditions can only be observed through case-by-case analysis (i.e., Fig. 11).

[Figure]

**Figure 11.** Validations of the Multi-textRG results on pure-CNN SAR SIE with different ice conditions. The first row shows the individual Sentinel-1 SAR HV grayscale images, the second row shows our U-Net predicted SIC maps ( ≥30% SIC is used to segment SIE), the third row shows the Multi-textRG detected SIE, and the fourth row shows the pure-CNN SAR SIE.

[Figure]

**Figure 1112.** Validation  metrics of the Net+Mulproduced ice extent: (a), (b) and (c) show the scatter values of the OA, FN and FP evaluation metrics validated on the first dataset, where the red and blue circles represent Landsat-8 and Sentinel-2 as validation data, respectively. (d), (e) and (f) show the scatter values of the OA, FN and FP evaluation metrics validated on the second dataset.

**References**

[1] Boulze, H., Korosov, A., & Brajard, J. (2020). Classification of Sea Ice Types in Sentinel-1 SAR Data Using Convolutional Neural Networks. Remote Sensing, 12

[2] Cristea, A., Johansson, A.M., Doulgeris, A.P., & Brekke, C. (2022). Automatic Detection of Low-Backscatter Targets in the Arctic Using Wide Swath Sentinel-1 Imagery. Ieee Journal of Selected Topics in Applied Earth Observations and Remote Sensing, 15, 8870-8883

[3] Guo, W., Itkin, P., Singha, S., Doulgeris, A.P., Johansson, M., & Spreen, G. (2023). Sea ice classification of TerraSAR-X ScanSAR images for the MOSAiC expedition incorporating per-class incidence angle dependency of image texture. The Cryosphere, 17, 1279-1297

[4] Jiang, M., Clausi, D.A., & Xu, L. (2022). Sea-Ice Mapping of RADARSAT-2 Imagery by Integrating Spatial Contexture With Textural Features. Ieee Journal of Selected Topics in Applied Earth Observations and Remote Sensing, 15, 7964-7977

[5] Kirillov, A., Mintun, E., Ravi, N., Mao, H., Rolland, C., Gustafson, L., Xiao, T., Whitehead, S., Berg, A.C., Lo, W.-Y., Dollar, P., & Girshick, R. (2023). Segment Anything. In (pp. 4015-4026)

[6] Leigh, S., Zhijie, W., & Clausi, D.A. (2014). Automated Ice–Water Classification Using Dual Polarization SAR Satellite Imagery. Ieee Transactions On Geoscience and Remote Sensing, 52, 5529-5539

[7] Li, X.-M., Sun, Y., & Zhang, Q. (2021). Extraction of Sea Ice Cover by Sentinel-1 SAR Based on Support Vector Machine With Unsupervised Generation of Training Data. Ieee Transactions On

Geoscience and Remote Sensing, 59, 3040-3053

[8] Liu, S., Zeng, Z., Ren, T., Li, F., Zhang, H., Yang, J., Jiang, Q., Li, C., Yang, J., Su, H., Zhu, J., & Zhang, L. (2024). Grounding DINO: Marrying DINO with Grounded Pre-Training for Open-Set Object Detection.

[9] Lohse, J., Doulgeris, A.P., & Dierking, W. (2020). Mapping sea-ice types from Sentinel-1 considering the surface-type dependent effect of incidence angle. Annals of Glaciology, 1-11

[10] Niehaus, H., Spreen, G., Birnbaum, G., Istomina, L., Jäkel, E., Linhardt, F., Neckel, N., Fuchs, N., Nicolaus, M., Sperzel, T., Tao, R., Webster, M., & Wright, N. (2023). Sea Ice Melt Pond Fraction Derived From Sentinel-2 Data: Along the MOSAiC Drift and Arctic-Wide. Geophysical Research Letters, 50

[11] Radford, A., Kim, J.W., Hallacy, C., Ramesh, A., Goh, G., Agarwal, S., Sastry, G., Askell, A., Mishkin, P., Clark, J., Krueger, G., & Sutskever, I. (2021). Learning Transferable Visual Models From Natural Language Supervision.

[12] Song, W., Li, M., Gao, W., Huang, D., Ma, Z., Liotta, A., & Perra, C. (2021). Automatic Sea-Ice Classification of SAR Images Based on Spatial and Temporal Features Learning. Ieee Transactions On Geoscience and Remote Sensing, 59, 9887-9901

[13] Stokholm, A., Buus-Hinkler, J., Wulf, T., Korosov, A., Saldo, R., Pedersen, L.T., Arthurs, D., Dragan, I., Modica, I., Pedro, J., Debien, A., Chen, X., Patel, M., Cantu, F.J.P., Turnes, J.N., Park, J., Xu, L., Scott, K.A., Clausi, D.A., Fang, Y., Jiang, M., Taleghanidoozdoozan, S., Brubacher, N.C., Soleymani, A., Gousseau, Z., Smaczny, M., Kowalski, P., Komorowski, J., Rijlaarsdam, D., van Rijn, J.N., Jakobsen, J., Rogers, M.S.J., Hughes, N., Zagon, T., Solberg, R., Longépé, N., & Kreiner, M.B. (2024). The AutoICE Challenge. The Cryosphere, 18, 3471-3494

[14] Stokholm, A., Wulf, T., Kucik, A., Saldo, R., Buus-Hinkler, J., & Hvidegaard, S.M. (2022). AI4SeaIce: Toward Solving Ambiguous SAR Textures in Convolutional Neural Networks for Automatic Sea Ice Concentration Charting. Ieee Transactions On Geoscience and Remote Sensing, 60, 1-13

[15] Sun, Y., Ye, Y., Wang, S., Liu, C., Chen, Z., & Cheng, X. (2023). Evaluation of the AMSR2 Ice Extent at the Arctic Sea Ice Edge Using an SAR-Based Ice Extent Product. Ieee Transactions On Geoscience and Remote Sensing, 61, 1-15

[16] Wang, Y.-R., & Li, X.-M. (2020). Arctic sea ice cover data from spaceborne SAR by deep learning. Earth System Science Data

[17] Zakhvatkina, N., Korosov, A., Muckenhuber, S., Sandven, S., & Babiker, M. (2017). Operational algorithm for ice–water classification on dual-polarized RADARSAT-2 images. The Cryosphere, 11, 33-46

[18] Zhang, J., Zhou, Z., Mai, G., Hu, M., Guan, Z., Li, S., & Mu, L. (2024a). Text2Seg: Remote Sensing Image Semantic Segmentation via Text-Guided V isual Foundation Models.

[19] Zhang, Z., Shen, H., Zhao, T., Guan, Z., Chen, B., Wang, Y., Jia, X., Cai, Y., Shang, Y., & Yin, J. (2025). Enhancing Ultra High Resolution Remote Sensing Imagery Analysis with ImageRAG. IEEE Geoscience and Remote Sensing Magazine, 14

[20] Zhang, Z., Zhao, T., Guo, Y., & Yin, J. (2024b). RS5M and GeoRSCLIP: A Large-Scale Vision-Language Dataset and a Large Vision-Language Model for Remote Sensing. Ieee Transactions On Geoscience and Remote Sensing, 62, 1-23